# VOILA: Evaluation of MLLMs for Perceptual Understanding and Analogical Reasoning

Nilay Yilmaz◇∗    Maitreya Patel◇    Yiran Lawrence Luo◇    Tejas Gokhale♡    Chitta Baral◇
Suren Jayasuriya◇    Yezhou Yang◇
◇Arizona State University    ♡University of Maryland, Baltimore County

## Abstract

Multimodal Large Language Models (MLLMs) have become a powerful tool for integrating visual and textual information. Despite their exceptional performance on visual understanding benchmarks, measuring their ability to reason abstractly across multiple images remains a significant challenge. To address this, we introduce VOILA, a large-scale, open-ended, dynamic benchmark designed to evaluate MLLMs' perceptual understanding and abstract relational reasoning. VOILA employs an analogical mapping approach in the visual domain, requiring models to generate an image that completes an analogy between two given image pairs, reference and application, without relying on predefined choices. Our experiments demonstrate that the analogical reasoning tasks in VOILA present a challenge to MLLMs. Through multi-step analysis, we reveal that current MLLMs struggle to comprehend inter-image relationships and exhibit limited capabilities in high-level relational reasoning. Notably, we observe that performance improves when following a multi-step strategy of least-to-most prompting. Comprehensive evaluations on open-source models and GPT-4o show that on text-based answers, the best accuracy for challenging scenarios is 13% (LLaMa 3.2) and even for simpler tasks is only 29% (GPT-4o), while human performance is significantly higher at 70% across both difficulty levels.

## 1    Introduction

Multimodal Large Language Models (MLLMs) have made remarkable strides in advancing human-level language processing and visual perception in tasks such as image captioning (Vinyals et al., 2015), visual question answering (Agrawal et al., 2016), object detection, and scene understanding (Bochkovskiy et al., 2020). While these advancements are promising, perceptual reasoning tasks such as relational and analogical reasoning, where models must infer and understand visual information, remain a significant challenge. Achieving human-level cognitive intelligence in these tasks demands greater attention and development.

According to Bloom's taxonomy of educational objectives, creation, rather than evaluation, requires the highest cognitive skills in the learning process (Bloom et al., 1956). However, many current multimodal reasoning tasks (Wang et al., 2024e; Plummer et al., 2016) rely on multiple-choice formats, where models select a solution from a predefined set. Although this approach provides insight into the learning and understanding capabilities of a model, we argue that it fails to reveal the model's ability to engage in high-level cognitive tasks involving the interpretation of visual context and abstract reasoning. To attain human-level cognitive intelligence, MLLMs must go beyond evaluating options; they must generate solutions for complex tasks that require advanced reasoning skills. Existing studies on open-ended visual reasoning tasks (Zellers et al., 2019; Zhang et al., 2019; Johnson et al., 2016), which assess MLLMs' cognitive capabilities, are limited in scope and do not fully explore these higher-order reasoning abilities.

In response to these challenges, we introduce VOILA: an open-ended reasoning benchmark designed to evaluate whether MLLMs possess vision-level understanding and relational reasoning capabilities. Our benchmark focuses on an *analogical reasoning* task which has been developed as

---

∗Corresponding author: nyilmaz3@asu.edu. Code and data: github.com/nlylmz/Voila

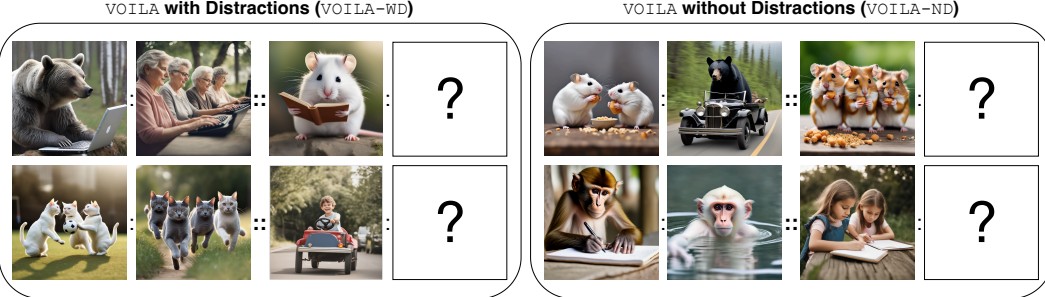

Figure 1: Examples of visual analogy questions from the VOILA benchmark with distractions (VOILA-WD) and without distractions (VOILA-ND). The aim is to generate an image that completes the analogy problem following perceptual and relational reasoning. Each visual analogy question has a specific rule configuration that leads to the answer. The questions in VOILA-WD benchmark can apply the distraction rule when no relational pattern exists between images. For the examples shown above, the answers that complete the VOILA-ND analogy by following the relation rules are *"two bears driving a car"* and *"two female children swimming"*. The answers to the VOILA-WD analogy are *"four [any subject] reading"* and *"two male children doing anything"*.

a cognitive assessment of problem-solving, decision-making, classification, and learning processes (Goswami, 1992). Analogical reasoning consists of diverse atomic abilities; perceptual understanding, mapping abstract relationships between visual contents (Gentner, 1983), and transferring relational patterns to novel cases. These integral sub-tasks are essential for achieving a coherent solution to the analogy problem. A key aspect of analogical reasoning is the transfer of knowledge from previously learned relations to new concepts. The task involves two image pairs: a reference pair and an application pair. The reference pair shares both visual and abstract contextual information. The goal is to infer the relationship and apply it to predict the unknown content in the application pair. A few examples are shown in Figure 1.

Our benchmark incorporates a multi-step reasoning process, which is crucial for analyzing where MLLMs encounter difficulties. This process allows for a detailed examination of the models' limitations about the key properties of the task. To introduce varying levels of difficulty, we created two sub-datasets: VOILA-WD and VOILA-ND. Our findings indicate that VOILA-WD presents a greater challenge than VOILA-ND due to the introduction of visual distractions where a certain property of the image (e.g. subject, number, task) is irrelevant to the analogical reasoning. Experimental results reveal that state-of-the-art MLLMs struggle on VOILA. Although GPT-4o reaches 79% accuracy in the description of the image and 43% precision in the identification of the relationship stages, it struggles to produce correct answers to complete the analogy. LLaMa 3.2 achieves the highest performance, attaining 13% accuracy in implementing the relationship stage on VOILA-WD. Interestingly, GPT-4o outperforms other models on VOILA-ND, achieving an accuracy of 29% in applying relationships. However, human performance significantly surpasses these results, achieving 71% and 69% accuracy on VOILA-WD and VOILA-ND, respectively.

We observe that least-to-most (L2M) prompting (Zhou et al., 2023) improves model accuracy compared to direct answering strategies. Input formats also influence model performance: using sequential images instead of a single image collage results in an average of 40% improvement. Additionally, the image generation step notably reduces model performance. We conduct an ablation study on VOILA-WD using GPT-4o, showing that even when ground truth information (image descriptions or relationships) is provided at each step, the model achieves only 17% accuracy on the applying relations step. Another ablation study on the VOILA-ND benchmark demonstrates that the model's performance improves by 27% when textual information is used instead of visual input. We describe our experimental setup and findings in Sections 4 and 5 in detail.

Our contributions and findings are summarized below:

• We present VOILA, a large-scale, open-ended benchmark to evaluate MLLMs' high-level visual reasoning capabilities. We introduce a method for the dynamic creation of extensive visual analogy questions utilizing text-to-image models.

Figure 2: Dataset creation pipeline of `VOILA`.

- We assess state-of-the-art MLLMs on `VOILA`, revealing a significant performance gap between humans (71%) and MLLMs (13%).
- We conduct comprehensive investigation of factors influencing performance: image format, prompting techniques, distraction rules, input information types, and provision of ground truths.

## 2 RELATED WORK

**Visual Analogical Reasoning.** Visual Analogical Reasoning is a complex task that requires models to recognize abstract similarities between visual contexts and underlying relational rules and to apply these rules to novel visual content to generate solutions. Raven's Progressive Matrices (RPMs) (Raven et al., 1938), introduced in 1938, are one of the earliest visual analogical reasoning tests, designed to assess human intelligence by completing a 3x3 matrix of visual patterns based on relational reasoning. Recent visual analogy benchmarks, such as VISALOGY (Sadeghi et al., 2015) and VASR (Bitton et al., 2022), offer valuable perspectives but have limitations. VISALOGY, which is not publicly available, focuses on attributes and actions in Google images with manual annotations, while VASR emphasizes general image understanding rather than feature-based relationships. In contrast, `VOILA` introduces a more complex and dynamic approach to visual analogical reasoning. It incorporates diverse subject relationships and rule-based structures and manipulates up to three properties at a time. Additionally, `VOILA` introduces distraction elements to increase task difficulty, requiring models to discover and filter out the irrelevant changes among properties while solving analogy questions. Unlike static datasets, `VOILA` allows the generation of over 6.4M distinct visual analogy scenarios across 14 subject types, 13 actions, and 4 numeric values by adjusting flexible property-rule configuration, offering a scalable and adaptable evaluation platform for MLLMs.

**Prompting.** Prompting strategies are critical to improving the performance of MLLMs, especially in complex reasoning tasks. Zero-shot prompting provides models with task instructions without examples, while few-shot prompting includes example-based prompts to guide the model toward the correct answer (Brown et al., 2020). Chain-of-Thought (CoT) prompting enhances reasoning by breaking down problems into sequential steps, allowing the model to tackle each sub-problem in a structured manner (Wei et al., 2022). Two common CoT approaches include zero-shot CoT (Kojima et al., 2022), which encourages step-by-step reasoning without examples, and few-shot CoT (Wei et al., 2022), which includes rationales or explanations to guide reasoning. Research has shown that few-shot CoT, by providing exemplar rationales, often outperforms zero-shot approaches (Zhang et al., 2024). Another method, Least-to-Most (L2M) prompting (Zhou et al., 2023), adopts a similar multi-phase structure as CoT but without the use of rationales. Instead, L2M progressively breaks down the task, using the solution to each sub-problem as input for the next. In `VOILA`, we extend the use of L2M to gain a deeper understanding of MLLMs' behavior across sub-tasks, from recognizing visual content to generating accurate images based on relational reasoning.

**Multimodal Reasoning Benchmarks.** Multimodal reasoning benchmarks have been instrumental in advancing the evaluation of MLLMs, integrating both textual and visual information to assess models' capabilities across a variety of domains. Domain-specific benchmarks like ScienceQA (Lu et al., 2022), A-OKVQA (Schwenk et al., 2022), Math-Vision (Wang et al., 2024b), and MMMU-Pro (Yue et al., 2024) focus on specialized knowledge, such as scientific, mathematical, and visual question-answering reasoning tasks. Meanwhile, benchmarks such as CompBench (Kil et al., 2024), MMRel (Nie et al., 2024), MARVEL (Jiang et al., 2024b), and ScanReason (Zhu et al., 2024) address more generalized multimodal relational reasoning. Several studies also focus on multi-step reasoning tasks, where models must process information sequentially, such as VisualCoT (Shao et al., 2024), LogicVista (Xiao et al., 2024), and VideoCoT (Wang et al., 2024d), while datasets

such as MuirBench (Wang et al., 2024a), MIRB (Zhao et al., 2024) and MANTIS-Eval (Jiang et al., 2024a) assess multiple image reasoning. Visual action planning (Gokhale et al., 2019) and visual procedural planning (Chang et al., 2020; Su et al., 2024) has also been explored to find relationships between pairs of images. However, VOILA distinguishes itself by focusing on high-order abstract relations and knowledge transfer across multiple images, challenging models not only to understand but also to generate both images and text while applying correct relational reasoning. These positions VOILA as a unique benchmark for testing MLLMs' higher-level cognitive abilities.

## 3 CONSTRUCTING THE VOILA BENCHMARK

The VOILA benchmark was designed to evaluate the abstract reasoning capabilities of MLLMs. This task challenges models to process perceptual information and apply relational reasoning by interpreting visual content from three given images to generate a fourth image according to a specified pattern. VOILA is a large-scale dataset that dynamically generates visual analogy questions based on demand and configuration. The dataset can generate over 6.4M questions, distributed across 19 unique structures and utilizing a total of 7,280 images which makes VOILA highly scalable and adaptable to various configurations. Figure 2 illustrates the dataset creation pipeline of VOILA.

### 3.1 DATASET CREATION PIPELINE

**Property Identification.** The VOILA dataset is generated using an image analogy framework $(A : A' :: B : B')$. Each of the first three images contains distinct properties that form the basis for the visual analogy questions. We identified three key properties: the number of subjects, subject type, and action. In the VOILA benchmark, each question $q_i$ includes three images $(I_1, I_2, I_3)$, with each image containing corresponding properties $(n_i, s_i, a_i) \in P$. A total of 14 subjects, 4 numbers, and 13 actions were used to create the image dataset. For further details regarding the categorical information of the property types, please refer to Appendix A.1.

**Rule Definition.** To structure each visual analogy question, four types of rules are applied in VOILA: Stable, Change, Arithmetic, and Distraction. The rules are assigned to the properties as outlined in Table 1, with each image containing rule-property pairs $I(r_i, p_i)$. Let $N$ symbolize the number of the subject property, $P_1$, $P_2$ and $P_3$ represent the same property (subject type or action) but different values. The rule patterns are defined as follows:

- **Stable:** The property value in the first image is the same as in the second image:

$$P_1 : P_1 :: P_2 : P_2.$$

- **Change:** The property value in the first image changes in the second image:

$$P_1 : P_2 :: P_1 : P_2.$$

- **Arithmetic:** The number of subjects changes by either increasing or decreasing from the first image to the second image. $N \geq 1$:

$$N_2 : N_4 :: N_1 : N_3 \rightarrow 4 - 2 = 2 :: 1 + 2 = 3.$$

- **Distraction:** The property values except the number of subjects are different in three images. There is no correlation among these values, so $P$ is a distraction. After applying increase or decrease changes, if $N \leq 0$, then $N$ is a distraction:

$$P_1 : P_2 :: P_3 : ANY . N_4 : N_1 :: N_2 : ANY \rightarrow 1 - 4 = -3 :: 2 - 3 = -1.$$

**Text Prompt and Image Generation.** To ensure that the relationships between images are easily recognizable, the images in VOILA must be clear and object-centered. We employ the open-source SDXL model, which generates high-quality images based on a simple text prompt structure that includes the number of subjects, subject types, and actions, for example *"Two dogs walking"*. Complex or overly detailed prompts can lead to incorrect image generation (Podell et al., 2023), so we maintain straightforward prompt structures. After generating text prompts, the images for the analogy questions are produced using the SDXL pipeline, with output resolution set to $1024 \times 1024$ and a guidance scale of 8, which controls the fidelity to the text prompt. For each prompt, 30 images are generated. Appendix A.2 provides examples of generated images and their text descriptions.

Table 1: `VOILA` contains analogies with three properties and four rules applied to these properties.

| Properties | Stable | Change | Arithmetic | Distraction |
|---|---|---|---|---|
| Action | ✓ | ✓ | ✗ | ✓ |
| Number of Subjects | ✓ | ✗ | ✓ | ✓ |
| Subject Type | ✓ | ✓ | ✗ | ✓ |

---

**Algorithm 1** Visual Analogy Generation

---

**Input:** Three property arrays: $numbers[]$, $subjects[]$, $actions[]$
**Output:** Question array: $Q[Image_0, Image_1, Image_2, Image_3]$
**Define** rules $R_y$ where $y = 1, 2, 3, 4$ {Stable, Changes, Arithmetic, Distraction}
**Define** analogy structures $A_i$ where $i = 1, 2, \ldots, 19$
Each structure $A_i$ contains variables $n_i$, $s_i$, $a_i$ and different rule configurations
**for** $i = 1$ to $19$ **do**
   $n_i \leftarrow \text{apply} R_y(numbers[])$ {Generate all possible permutations of properties}
   $s_i \leftarrow \text{apply} R_y(subjects[])$
   $a_i \leftarrow \text{apply} R_y(actions[])$
   $combinations[] \leftarrow \text{combineProperties}(n_i, s_i, a_i, count)$ {Generate random selections with balanced distribution}
   **for** $n, s, a$ in $combinations[]$ **do**
      **for** $x = 0$ to $3$ **do**
         $Image_x \leftarrow \text{findImage}(n[x], s[x], a[x], index)$ {Find images that matches requested properties. Use the index to provide a variety of images.}
      **end for**
      $Q \leftarrow (Image_0, Image_1, Image_2, Image_3)$
   **end for**
**end for**
**End**

---

**Data Cleaning.** It is essential to verify the alignment between the text prompts and the generated images. Some images may not correspond correctly to their prompts (see Appendix A.3), making them unsuitable for visual analogy construction. These images were manually filtered, and from this process, we retained 10 images per prompt, resulting in a total of 7,280 diverse images.

**Building Image Analogies.** To construct visual analogy questions, we combine image features using predetermined rules. Each question follows a specific rule-property configuration, with the number of properties undergoing changes determining the dataset configuration. Since `VOILA` includes three properties, it proposes three different analogy structures. The number of cases required for each structure is calculated by pairing unchanged properties with unmodified rules. Given that Arithmetic and Change rules modify properties, the remaining two rules are used for the process.

$$C = r^{n-p} \cdot \frac{n!}{p! \cdot (n-p)!},\tag{1}$$

where $r$ is the number of rules (excluding Arithmetic and Change), $n$ is the total number of properties, and $p$ indicates the number of changed properties. This yields 19 unique cases for three configurations: 12 cases for 1 property change, 6 cases for 2 property changes, and 1 case where all properties change. Appendix A.5 details the possible cases and assigned rules for each property.

To create a dynamic `VOILA` benchmark, we propose the Visual Analogy Generation strategy (see Algorithm 1), which takes input properties, structure, and rule configurations and outputs the corresponding image sequence for each `VOILA` question. Each structure is assigned a property-rule configuration, and the number of distinct analogy questions for each structure is calculated based on rule-property permutations, including dual permutations for the Stable and Change rules, and length-3 permutations for the Distraction rule. Unique number combinations for Arithmetic and Distraction rules are manually calculated to eliminate redundancy (see Appendix A.5 for details).

In `VOILA`, the analogy questions are equally distributed among the 19 structures, ensuring a distinct combination of properties for each question. Images can be reused across different configurations, preserving uniqueness while distributing image usage evenly.

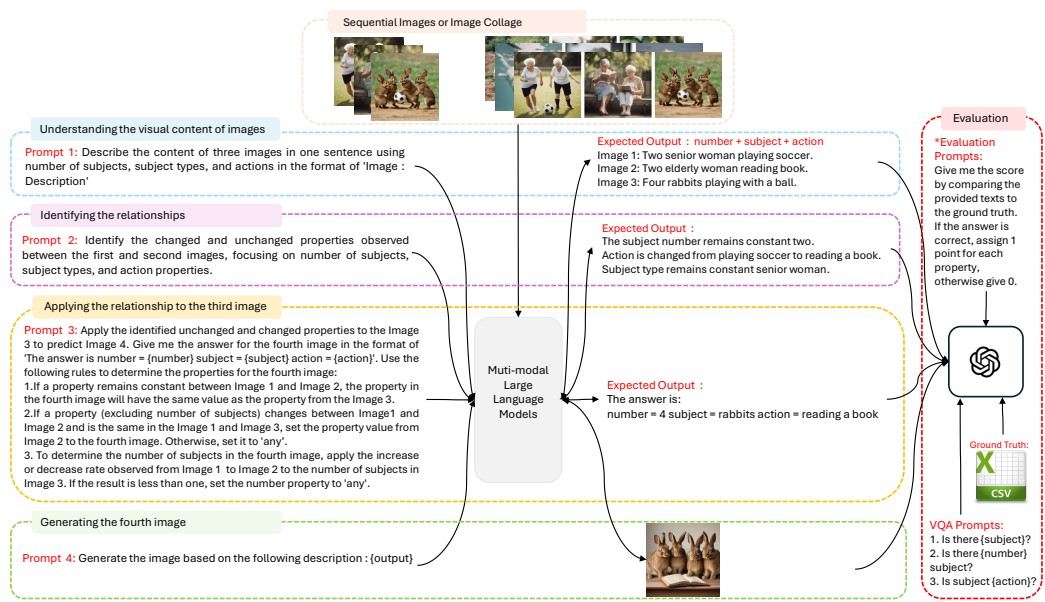

Figure 3: `VOILA` multi-step reasoning and evaluation pipeline. The top section illustrates two visual input formats. The left side of the MLLMs connection displays the four primary tasks along with their corresponding prompts, while the right side presents the expected outcomes for each task. The results are scored in the evaluation stage utilizing GPT-4o and ground truths.

## 4 EXPERIMENTAL SETUP

**`VOILA` Test Dataset.** For evaluating MLLMs, the `VOILA` benchmark is split into two distinct test datasets: `VOILA-WD`, which includes Distraction rules, and `VOILA-ND`, which excludes them. `VOILA-WD` consists of 10K unique questions, applying all four rules across 19 structures, while `VOILA-ND` comprises 3.6K questions with seven configurations and three rules (see Appendix A.4 for details). Each dataset contains 527 questions per configuration, with 728 images annotated with image contents, relationship explanations, and descriptions of the requested image.

**Models.** We evaluated several baseline models on both `VOILA-WD` and `VOILA-ND`: GPT-4o (Achiam et al., 2023), Emu2-37B (Sun et al., 2024), Qwen2-VL-7B-Instruct (Wang et al., 2024c), CogVLM2-19B (Hong et al., 2024), SEED-LLaMA-8B (Ge et al., 2023), Llama 3.2-11B, and MolmoE-7B (Deitke et al., 2024). Models incapable of producing visual output were excluded from the image generation task.

**Image Input.** Given that some baseline models, such as CogVLM2, do not support multiple image inputs, we implemented two visual input formats: a sequential presentation of the three images and an image collage combining the three images into a single visual representation.

**Prompting.** We applied the Least-to-Most (L2M) prompting strategy (Zhou et al., 2023) and manually decomposed the visual analogy task into four sub-problems: (1) understanding the visual content, (2) identifying relationships between images, (3) applying those relationships to the third image, and (4) generating the content of the fourth image. Instead of using sub-questions, we employed sub-instructions, asking the models to solve each sub-task sequentially, with the previous answer appended to the next problem. This structured reasoning process allowed us to evaluate performance at each sub-task. We tested various prompts on the baseline models using both L2M and direct answer approaches. The prompts used for multi-step reasoning and direct answering are detailed in Appendix C.1.

**Evaluation.** Performance on `VOILA` is assessed at each step based on correct property prediction. Using GPT-4o and four distinct text prompts, we scored model responses for each step, see Figure

Table 2: Evaluation results of `VOILA` using least-to-most (L2M) (Zhou et al., 2023) prompting.

| Method | VOILA-WD (VOILA w/ Distraction) | | | VOILA-ND (VOILA w/o Distraction) | | |
|---|---|---|---|---|---|---|
| | *Describing Images* | *Identifying Relations* | *Applying Relationship* | *Describing Images* | *Identifying Relations* | *Applying Relationship* |
| Human (AMTurks) | - | - | **71.36** | - | - | **69.69** |
| **Image Collage** | | | | | | |
| COG-VLM2 | 56.05 | 14.54 | 0.41 | 57.93 | 19.13 | 6.39 |
| Qwen-VL2-Chat | 48.41 | 12.25 | 0.52 | 44.80 | 9.87 | 3.77 |
| GPT-4o | 65.37 | 24.93 | 3.94 | 64.45 | 25.00 | 19.43 |
| LLaMa 3.2 | 68.28 | 29.00 | **13.16** | 67.88 | 24.45 | 6.83 |
| **Three Separate Images** | | | | | | |
| MolmoE | 7.76 | 0.65 | 0.08 | 8.34 | 0.38 | 1.00 |
| Qwen-VL2-Chat | 77.80 | 20.58 | 0.85 | 75.80 | 21.20 | 5.20 |
| GPT-4o | **78.94** | **42.79** | 6.44 | **78.53** | **38.60** | **29.03** |

3. In the first phase, the models' ability to understand the visual content is evaluated. Given three images $I_{i_j}$ and their properties $P_{i_j}$ and ground truth descriptions $G_i$, the score in this phase is calculated as:

$$f(Q_i) = \begin{cases} 1 & \text{if } Q_i(P_{i_j}(n, s, a), I_{i_j}) = Q_i(G_i, I_{i_j}) \text{ and } j \in [0, 2] \\ 0 & \text{otherwise.} \end{cases} \quad (2)$$

This scoring strategy is applied across the first three phases, where models are evaluated on whether they correctly identify the properties of the images. In the second phase, we assess the models' ability to extract relationships between the images. In the third phase, we evaluate the application of these relationships to a new domain. In the final step, the model-generated images are assessed using a VQA-style approach. For each property, we generated three questions based on the ground truth text and used GPT-4o to answer these questions in relation to the generated image. A similar property-based scoring method is applied to evaluate the generated images. Detailed evaluation prompts for each step are provided in Appendix B.12.

**Human Performance.** We also evaluated human performance on the visual analogy task using the Amazon Mechanical Turk (MTurk) platform. A total of 440 distinct analogy questions, incorporating different rule configurations, were presented to participants. Instruction sets and example questions, with and without Distraction rules, were provided during the experiment. Human participants were tasked with predicting the properties of the missing fourth image. The results were collected and evaluated against the ground truth text. Additional details on the MTurk human evaluation study are available in Appendix D.1.

## 5 EXPERIMENTAL RESULTS

### 5.1 MAIN RESULTS

We evaluated the high-level reasoning abilities of state-of-the-art MLLMs on the `VOILA` benchmark. Table 2 presents the step-by-step accuracy of the models on both `VOILA-WD` and `VOILA-ND`. Although both GPT-4o and Qwen-VL2 achieved peak performance with an average accuracy of 78% during the image description stage, GPT-4o distinguished itself with exceptional results in relational reasoning by reaching an average precision of 40% in both datasets. In the application relationship step, while GPT-4o outperformed other models by 29%, emerging as the top performer on `VOILA-ND`, LLaMa 3.2 took the lead on `VOILA-WD` with a 13% accuracy rate, surpassing GPT-4o by 6.7%. This result shows LLaMa 3.2's enhanced ability to identify distractions compared to other models. However, human participants still significantly outperformed all MLLMs, particularly in understanding relationships and making inferences. The performance gap between human participants and the top-performing models—LLaMa 3.2 on `VOILA-WD` and GPT-4o on `VOILA-ND`—equals approximately 58% and 40%, respectively.

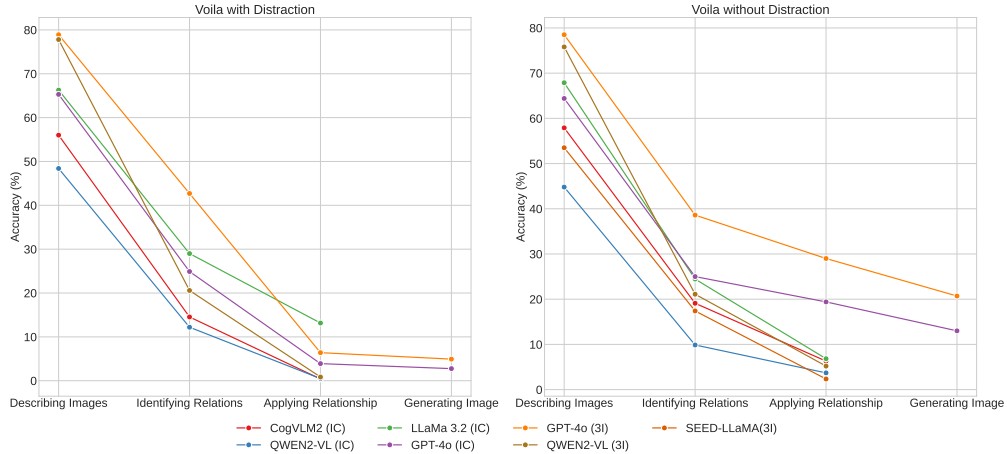

Figure 4: Accuracy of `VOILA-WD` and `VOILA-ND` at each step, respectively.

Model accuracy notably declines after each reasoning step, illustrating the growing difficulty as tasks become more complex. Figure 4 shows the accuracy of models across the four reasoning stages on `VOILA-WD` and `VOILA-ND`. While most models achieve over 50% accuracy in the initial image understanding stage, their performance drops sharply in the second stage, where they are required to interpret relationships. This decline continues in the third stage, where models apply these relationships to generate inferences for the third image. Due to the limited image generation capabilities of baseline models, we evaluated this step only on GPT-4o, Seed-LLaMA, and Emu-2. Across both datasets, performance dropped at the image generation stage (see Appendix B.2). These results demonstrate that current MLLMs struggle with relational understanding and their inference accuracy. Additional details on successful and failed cases are available in Appendix B.10.

## 5.2 PROPERTY SUCCESS OF MODELS

To further analyze how baseline models understand and predict properties, we evaluated model performance based on three properties: the number of subjects, subject type, and action, across each reasoning step on `VOILA-WD` and `VOILA-ND`. Figure 5 highlights the property-based accuracy of four models that perform best on `VOILA-WD`. Each model exhibited different strengths in identifying properties at each step. QWEN2-VL, for instance, exhibited strong capabilities in identifying numbers and subject types during the initial two steps, but its performance notably decreased when applying relationships, particularly for these properties similar to CogVLM-2. GPT-4o maintained high accuracy in predicting numbers during the first and second stages but experienced a significant decrease of 60% in the relation application phase on `VOILA-WD`, followed by an additional 6% decline in the image generation step. Although QWEN2-VL and GPT-4o performed best during the first stage, QWEN2-VL struggled more than GPT-4o to identify the relationships. LLaMa 3.2 achieves the most balanced performance across all categories, maintaining relatively high accuracy in the complex task of relationship application. While transferring the relationship step is the most challenging part for CogVLM-2, QWEN2-VL, and GPT-4o, LLaMa 3.2 struggles more with identifying relationships on `VOILA-WD`. Further property analysis for `VOILA-WD` is provided in Appendix B.3.

## 5.3 L2M VS DIRECT ANSWER

We conducted experiments comparing Least-to-Most (L2M) prompting with direct answering approaches on `VOILA-WD` and `VOILA-ND`. Table 3 summarizes the results, showing that L2M prompting consistently improves model performance by breaking down the visual analogy task into sub-problems. This approach leads to higher accuracy in both settings, with a particularly strong impact on `VOILA-WD`, which involves more complex rule configurations. These findings suggest that L2M prompting enhances the reasoning process by encouraging models to solve each part of the problem incrementally. Additional results on direct answering are available in Appendix B.1.

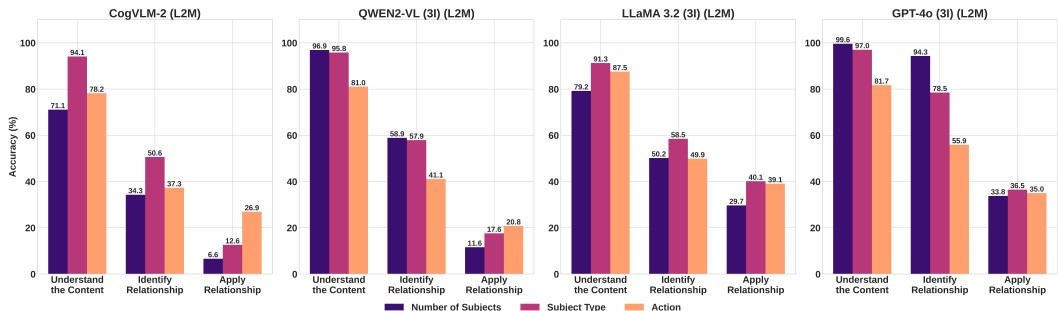

Figure 5: Accuracy of baseline MLLMs regarding properties at each step on `VOILA-WD`.

Table 3: Accuracy of the models in the third step using L2M and direct answering.

| | VOILA-WD | | VOILA-ND | |
|---|---|---|---|---|
| Models | L2M | Direct Answer | L2M | Direct Answer |
| COG-VLM2 (IC) | 0.41 | 0.18 | 6.39 | 3.57 |
| GPT-4o (3I) | 6.44 | 0.9 | 29.03 | 16.94 |
| Seed LLama 14B (3I) | 2.99 | 1.35 | 2.35 | 2.03 |

Table 4: Accuracy of the models in the third step using L2M on image collage and sequential image settings.

| | VOILA-WD | | VOILA-ND | |
|---|---|---|---|---|
| Models | Image Collage | Sequential Images | Image Collage | Sequential Images |
| QWEN-VL2 (L2M) | 0.52 | 0.85 | 3.77 | 6.8 |
| GPT-4o (L2M) | 3.94 | 6.44 | 19.43 | 29.03 |

## 5.4 IMAGE COLLAGE VS SEQUENTIAL IMAGE

We examined the impact of input formats—image collage versus sequential images—on model performance in `VOILA-WD` and `VOILA-ND`. Table 4 compares the results for Qwen2-VL and GPT-4o, which accept both input formats. Our findings indicate that input configuration has a significant effect on visual perception: performance dropped by approximately 40% when models were presented with image collages compared to sequential images. This pattern of performance degradation was consistent across both `VOILA-WD` and `VOILA-ND`, highlighting that models struggle with visual analogy tasks when images are combined in a collage format, which is counterintuitive and can be attributed to the image resolution constraints. However, our additional study with LLaVA-OneVision (Li et al., 2024) detailed in Appendix B.6, demonstrates that the AnyRes approach enhances the interpretation of collaged images and helps to reduce the performance differences.

## 5.5 `VOILA-WD` VS `VOILA-ND`

According to the accuracy outcomes of models on both `VOILA-WD` and `VOILA-ND` benchmarks, all models, excluding LLaMa 3.2, perform better addressing the `VOILA-ND` questions, see Figure 6. The accuracy of best-performer GPT-4o, dropped by 22% when solving `VOILA-WD` questions. The results demonstrate that implementing the Distraction rule increases the difficulty level of the `VOILA` benchmark and proves that `VOILA-WD` introduces more complex challenges compared to `VOILA-ND`. We also analyzed the rule-based performances of LLaMa 3.2 and discovered that it applies the Distraction rule better than other rules (Arithmetic and Stable), particularly in number property which explains why it achieves better results on the `VOILA-WD` unlike other models.

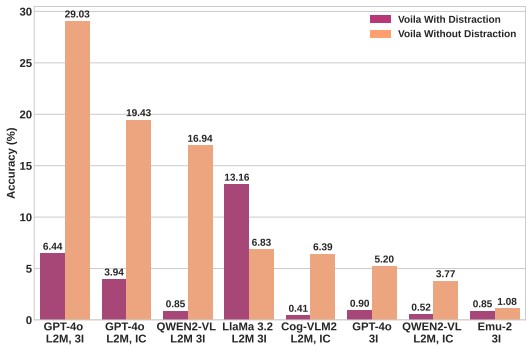

Figure 6: Performance of the top-8 models on `VOILA-ND` and `VOILA-WD`.

## 5.6 ERROR ANALYSIS

To assess GPT-4o's performance on the evaluation task and to quantify the differences between human and GPT-4o evaluations, we examined 50 visual analogy questions and 180 responses, 30 corresponding to the image generation step, answered by various models using diverse rule configurations including the distraction rule. The results show that GPT-4o has an error rate of up to 10% per step, and its challenges in recognizing relationships do not influence the benchmark assessment procedure. Additional details of the analysis are available in Appendix B.5

## 6 ABLATION STUDY

### 6.1 MODEL PERFORMANCE WITH ACCESS TO GROUND TRUTH INFORMATION

To better understand model behavior at each reasoning step, we conducted an ablation study on GPT-4o by providing ground truth inputs starting from the second phase. This study aims to evaluate the model's reasoning abilities under ideal conditions. Using L2M prompting on the `VOILA-WD` benchmark, we provided GPT-4o with ground truth image descriptions at phase 2. The model's performance in identifying relationships increased significantly, reaching 97%, indicating that GPT-4o can effectively analyze changes in text descriptions when given correct inputs. However, when we provided the model with ground truth relationships in the third step, its performance dropped dramatically to 17%, well below human performance (71%). These results suggest that GPT-4o struggles to apply known relationships to new visuals, revealing limited reasoning in practical inference.

### 6.2 HOW DOES VISUAL INFORMATION AFFECT PERFORMANCE?

We also investigated how visual versus textual information affects model performance on the analogy task by conducting an ablation study on `VOILA-ND`. In this experiment, we used a direct answering prompt without any explanation of the rules. Two experiments were conducted with GPT-4o, one using three sequential images as input and the other using text descriptions of those images. When processing image data, GPT-4o achieved an accuracy of 22%, while with textual input, its accuracy rose to 49%. These results highlight the importance of input format in MLLM performance, exposing a gap between visual and textual reasoning abilities.

We also analyze the results based on the number of correctly answered questions on `VOILA-ND`. Figure 7 demonstrates the rule-based performance comparison of models that accept visual and textual information. The outcomes of both models show that the Arithmetic rule included in rule numbers 6 and 7 influences GPT-4o's performance. The models present better accuracy when Stable or Change rules are applied to the number property. We also observed that models achieve the lowest accuracy in predicting the properties in rule 7 where all properties in question change at a time.

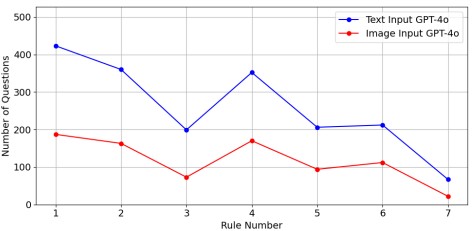

Figure 7: Correct number of questions in `VOILA-ND`.

## 7 CONCLUSION

We introduced `VOILA`, a large-scale, open-ended, and dynamic reasoning benchmark designed to evaluate the visual understanding and analogical reasoning capabilities of state-of-the-art MLLMs. `VOILA` comprises two sub-tasks: the more complex `VOILA-WD` and the simpler `VOILA-ND`. Our evaluations revealed that humans outperform the best-performing models by a substantial margin of 58% on `VOILA-WD` and 40% on `VOILA-ND`. These results demonstrate that current MLLMs not only struggle to generate accurate visual or textual outputs but also lack the ability to recognize and apply relational reasoning across images. The significant performance gap between humans and MLLMs underscores the limitations of current models in higher-level cognitive tasks. We anticipate that `VOILA` will serve as a rigorous benchmark for advancing MLLMs, systematically evaluating their ability to tackle complex reasoning tasks that demand human-like intelligence.

## ACKNOWLEDGMENTS

NY is supported by the Republic of Türkiye Ministry of National Education. MP, CB, and YY are supported by US NSF RI grant #2132724. TG was supported by the SURFF award from UMBC ORCA. We thank the NSF NAIRR initiative, the Research Computing (RC) at Arizona State University (ASU), and cr8dl.ai for their generous support in providing computing resources. The views and opinions of the authors expressed herein do not necessarily state or reflect those of the funding agencies and employers.

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

## A  THE VOILA DATASET

### A.1  PROPERTY DETAILS

We select three properties to utilize in the VOILA: subject types, number of subjects and actions. In total 14 distinct subject types used in the images: cat, dog, fox, hamster, rabbit wolf, bear, and monkey, male child, female child, man, woman, senior man, and senior woman. Also, each image includes one subject type. The numbers selected for the dataset are 1, 2, 3 and 4. We want to restrict the numbers to avoid creating more complex configurations. Action property includes 13 physical activities: playing soccer, driving a car, ice-skating, walking, swimming, jumping, typing, writing, digging a hole, carrying something, reading, running, and eating food.

### A.2  GENERATED IMAGES BY SDXL

Utilizing the SDXL text-to-image model, we generated 30 images for each prompt using various properties. Some of the generated images are provided in Figures 8, 9, 10, and 11 with their text prompts.

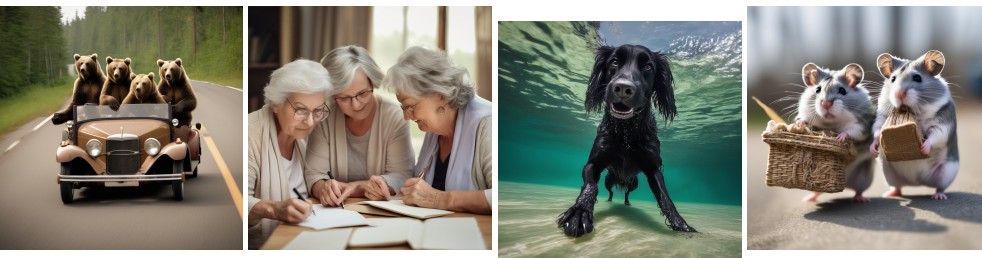

Figure 8:
Four brown bears
driving a car

Figure 9:
Three senior women
writing

Figure 10:
One dog swimming

Figure 11:
Two hamsters carrying
something

### A.3  IMAGE CLEANING

Some of the images generated by the model obtain some problems like object hallucinations and not depicting the action clearly. To eliminate these failure cases, we filter the images manually. Figure 12 shows some of the faulty images generated by the model.

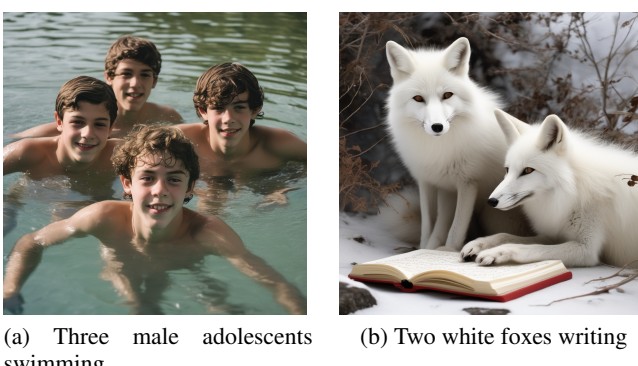

(a) Three male adolescents swimming

(b) Two white foxes writing

Figure 12: Unmatched images with text prompt.

### A.4  VOILA STRUCTURE

The configuration of VOILA benchmarks is determined by the rules and the number of properties. Table 5 shows all possible rule settings for three property-based cases using all rules that represent

the configuration of `VOILA-WD`. On the other hand, Table 6 shows seven distinct cases excluding the Distraction rule.

Table 5: In total 19 cases are required to generate visual analogy questions in `VOILA-WD`.

| Rule | Action | Subject Type | Number of Subject |
|------|--------|--------------|-------------------|
| 1 | Change | Stable | Stable |
| 2 | Change | Stable | Distraction |
| 3 | Change | Distraction | Stable |
| 4 | Change | Distraction | Distraction |
| 5 | Stable | Change | Stable |
| 6 | Stable | Change | Distraction |
| 7 | Distraction | Change | Stable |
| 8 | Distraction | Change | Distraction |
| 9 | Stable | Stable | Arithmetic |
| 10 | Stable | Distraction | Arithmetic |
| 11 | Distraction | Stable | Arithmetic |
| 12 | Distraction | Distraction | Arithmetic |
| 13 | Change | Change | Stable |
| 14 | Change | Change | Distraction |
| 15 | Stable | Change | Arithmetic |
| 16 | Distraction | Change | Arithmetic |
| 17 | Change | Stable | Arithmetic |
| 18 | Change | Distraction | Arithmetic |
| 19 | Change | Change | Arithmetic |

Table 6: In total 7 cases are required to generate visual analogy questions in `VOILA-ND`.

| Rule | Action | Subject Type | Number of Subject |
|------|--------|--------------|-------------------|
| 1 | Change | Stable | Stable |
| 2 | Stable | Change | Stable |
| 3 | Stable | Stable | Arithmetic |
| 4 | Change | Change | Stable |
| 5 | Stable | Change | Arithmetic |
| 6 | Change | Stable | Arithmetic |
| 7 | Change | Change | Arithmetic |

Table 7: Number of different cases created by rules.

| Properties | Stable | Change | Arithmetic | Distraction |
|------------|--------|--------|------------|-------------|
| Action | 156 | 156 | - | 22308 |
| Number of Subjects | 12 | - | 16 | 24 |
| Subject Type | 182 | 182 | - | 30576 |

## A.5 CASE NUMBERS

To calculate how many different cases we can generate using properties, we implement dual permutations for the Stable and Change rules and a permutation of three for the Distraction rules. Potential cases of the number of subjects are manually calculated. Table 7 shows the number of various possibilities for each property after assigning the rules. For 14 subject types, the Stable and Change rules generate 182 cases that can be used in the visual analogy questions. Since the Distraction number has a large amount of data variation, I fixed the generated number of data according to the permutation results of Changes and Stable rules. Utilizing these variations, we can create more than 6.5M analogy questions.

## B  EXPERIMENT RESULTS

### B.1  DIRECT ANSWERING

The reasoning workflow is applied with final determined prompts progressively for each model. We tested the direct answering approach for some benchmark models on both `VOILA-WD` and `VOILA-ND`. The models are tested on a zero-shot prompting approach without a multi-step process. The steps of image description and understanding the relations are skipped in that study to directly receive the answer from the model. The results are provided in Table 8.

Table 8: Evaluation results of `VOILA` using direct answering on *Applying Relationship*. In this setting, there are no immediate results for *Describing Images* and *Identifying Relations*.

| Method | VOILA-WD | VOILA-ND |
|---|---|---|
| Human (AMTurks) | **71.36** | **69.69** |
| **Image Collage** | | |
| Seed LLama 14B | 2.14 | 0.76 |
| **Three Separate Images** | | |
| GPT-4o | 0.90 | 16.94 |
| Emu-2 | 0.85 | 1.08 |
| Seed LLama 14B | 1.35 | 2.03 |

Table 9: Evaluation results of `VOILA` on *Generating Images*. L2M = least-to-most prompting, DA = direct answering.

| Method | VOILA-WD | VOILA-ND |
|---|---|---|
| **Image Collage** | | |
| GPT-4o (L2M) | 2.76 | 13.01 |
| **Three Separate Images** | | |
| GPT-4o (L2M) | **4.93** | 20.76 |
| GPT-4o (DA) | 0.67 | 11.3 |
| Seed LLama 14B (DA) | 0.11 | - |
| Emu-2 (DA) | 0.03 | 0.11 |

### B.2  IMAGE GENERATION

Image generation is the last step of the visual analogy questions. To answer the question correctly, MLLMs that are capable of creating visual outputs, need to generate the accurate images aligned with the ground truths. We tested GPT-4o and Seed LLaMa on `VOILA-WD` and `VOILA-ND` datasets to evaluate their performance of image generation after the relational inference step. Since `VOILA-ND` has less challenging questions, the models perform better on `VOILA-ND` rather than `VOILA-WD` which obtains some distraction rules. Since the models struggle to maintain their performance until the last step, the accuracy of the image generation performance is below 5% of the best-performing model (GPT-4o) on `VOILA-WD`. The more detailed results are provided in Table 9.

### B.3  VOILA-WD PROPERTY-BASED EVALUATION

We comprehensively analyze the performance of the baseline models against the accuracy of the properties of each step. Table 10 and Figure 5 show the performance of various models in `VOILA-WD` with respect to different configurations such as using L2M or directly answering and entering an image collage or three sequential images. The results demonstrate that GPT-4o performs better than other models in understanding and identifying relations across all properties in visual content on `VOILA-WD`. However, LLaMa 3.2 shows more thriving performance in applying relationships on subject types and actions.

### B.4  VOILA-ND PROPERTY-BASED EVALUATION

We evaluated models regarding property-based performance at each step on `VOILA-ND` under different configurations, including the use of L2M versus direct answering, as well as the input format of an image collage compared to three sequential images. As the distraction rule affects the application relationships stage, the model performances at the initial two stages show a similar pattern as in the `VOILA-WD` dataset. The results provided in Table 11 and Figure 13 demonstrate that all models, excluding LLaMa 3.2, achieved better performance on `VOILA-ND` and increased the accuracy of property predictions in the final step. In contrast, LLaMa 3.2 exhibited a decrease in performance at the last stage, particularly in the application of number relations. Notably, GPT-4o outperformed other models in applying number and subject relations.

Table 10: Performance of various models on different steps `VOILA-WD`. Model names: IC = image collage, 3I = three sequential images, L2M = least-to-most prompting.

| Model | Describing Images | | | Identifying Relations | | | Applying Relationship | | | Generating Image | | |
|---|---|---|---|---|---|---|---|---|---|---|---|---|
| | Number | Subject | Action | Number | Subject | Action | Number | Subject | Action | Number | Subject | Action |
| Humans (MTurk) | - | - | - | - | - | - | **93.6** | **82.3** | **91.4** | - | - | - |
| CogVLM2 (L2M) | 71.1 | 94.1 | 78.2 | 34.3 | 50.6 | 37.3 | 6.6 | 12.6 | 26.9 | - | - | - |
| LLaVa (L2M) | 75.4 | 87.8 | 67.5 | 30.7 | 32.6 | 27.8 | 31.2 | 30.0 | 32.9 | - | - | - |
| QWEN2-VL (L2M) | 62.0 | 89.2 | 76.1 | 32.2 | 53.6 | 38.8 | 3.16 | 21.8 | 1.9 | - | - | - |
| QWEN2-VL (3I) (L2M) | 96.9 | 95.8 | 81.0 | 58.9 | 57.9 | 41.1 | 11.6 | 17.6 | 20.8 | - | - | - |
| MolmoE (3I) (L2M) | 14.57 | 32.22 | 22.18 | 6.61 | 10.92 | 8.02 | 5.62 | 10.06 | 10.63 | - | - | - |
| LLaMa 3.2 (IC) (L2M) | 79.2 | 91.3 | 87.6 | 50.2 | 58.5 | 49.9 | 29.7 | **40.1** | **39.1** | - | - | - |
| SEED-LLaMa (3I) (L2M) | 89.1 | 87.3 | **93.7** | 48.8 | 45.8 | 48.0 | 12.1 | 19.7 | 24.8 | - | - | - |
| GPT-4o (L2M) | 86.0 | 95.9 | 78.8 | 68.1 | 70.9 | 46.5 | 25.4 | 33.8 | 33.2 | 3.4 | 3.7 | 3.2 |
| GPT-4o (3I)(L2M) | **99.6** | **97.0** | 81.7 | **94.3** | **78.5** | **55.9** | 33.8 | 36.5 | 35.0 | **5.6** | **6.1** | **5.6** |
| GPT-4o (3I) | - | - | - | - | - | - | 10.9 | 13.7 | 27.5 | 0.8 | 0.8 | 0.8 |
| SEED-LLaMa (3I) | - | - | - | - | - | - | 5.1 | 16.9 | 25.8 | 1.1 | 0.7 | 0.7 |
| Emu-2 | - | - | - | - | - | - | 3.0 | 6.1 | 12.5 | 0.2 | 0.2 | 0.1 |
| CogVLM2 | - | - | - | - | - | - | 7.8 | 8.2 | 23.7 | - | - | - |
| SEED-LLaMa | - | - | - | - | - | - | 9.2 | 30.2 | 6.8 | - | - | - |

Table 11: Performance of various models on different steps `VOILA-ND`. Model names: IC = image collage, 3I = three separate images, L2M = least-to-most prompting.

| Model | Describing Images | | | Identifying Relations | | | Applying Relationship | | | Generating Image | | |
|---|---|---|---|---|---|---|---|---|---|---|---|---|
| | Number | Subject | Action | Number | Subject | Action | Number | Subject | Action | Number | Subject | Action |
| Humans (MTurk) | - | - | - | - | - | - | **92.4** | **85.4** | **91** | - | - | - |
| CogVLM2 (L2M) | 72.9 | 95 | 79 | 45.7 | 59.1 | 45.8 | 23.8 | 37.1 | 21.4 | - | - | - |
| QWEN2-VL (L2M) | 56.8 | 86.7 | 74.9 | 30.7 | 54 | 34.9 | 19.8 | 25.8 | 21.7 | - | - | - |
| QWEN2-VL (3I)(L2M) | 96.9 | 95.4 | 81 | 61.9 | 57.5 | 42.4 | 15.8 | 27 | 27.2 | - | - | - |
| MolmoE (3I) (L2M) | 15.5 | 37.6 | 26.9 | 6.23 | 10.6 | 8.5 | 8.9 | 22 | 14.2 | - | - | - |
| LLaMa 3.2 (IC) (L2M) | 77.8 | 90.9 | **88.4** | 50.9 | 60.9 | 49.2 | 15.1 | 50.1 | 48.3 | - | - | - |
| SEED-LLaMa (3I)(L2M) | 77.9 | 86.8 | 73.1 | 48.1 | 54.9 | 37.4 | 7.2 | 21.8 | 27.2 | - | - | - |
| GPT-4o (L2M) | 84.8 | 95.9 | 78.9 | 69.2 | 74.3 | 45.8 | 45.8 | 61.2 | 44.9 | 15.9 | 17.9 | 16.9 |
| GPT-4o(3I)(L2M) | **99.6** | **96.6** | 81.5 | **94** | **77.6** | **51.4** | **66.4** | **65.6** | **49.1** | **24.3** | **27.2** | **26.1** |
| GPT-4o(3I) | - | - | - | - | - | - | 28.9 | 45.5 | 18.7 | 14.4 | 15.1 | 14.6 |
| Emu-2 | - | - | - | - | - | - | 6.2 | 8.7 | 13.1 | 0.4 | 0.7 | 0.3 |
| CogVLM2 | - | - | - | - | - | - | 13 | 19.9 | 17.9 | - | - | - |
| SEED-LLaMa (3I) | - | - | - | - | - | - | 6.8 | 7.8 | 24.3 | - | - | - |
| SEED-LLaMa | - | - | - | - | - | - | 4.3 | 11.9 | 13.1 | - | - | - |

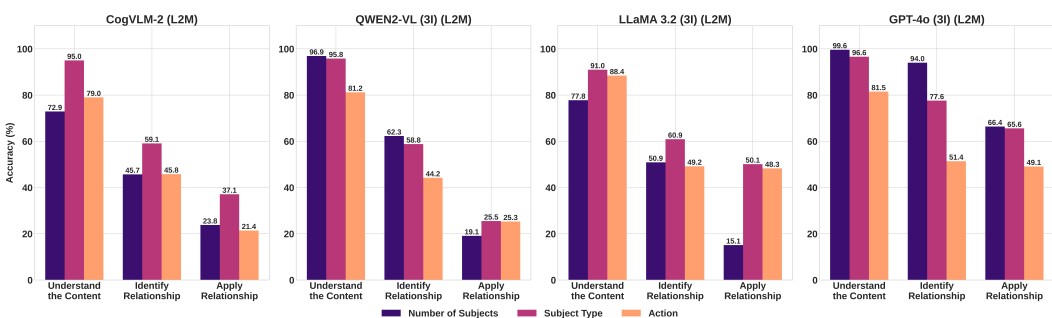

Figure 13: Property performance of models at each step on `VOILA-ND`

### B.5 GPT-4O EVALUATION PERFORMANCE

We measured the gap between human and GPT-4 evaluations utilizing confusion tables for each step including all attributes. In total 50 visual analogy questions and 180 responses were evaluated.

- True Positive (TP): The number of cases where both the human and the GPT-4o agree that the response is correct.
- False Negative (FN): The number of cases where the human expresses the answer is correct, but the GPT-4o says it is incorrect.

- False Positive (FP): The number of cases where the human says the answer is incorrect, but the GPT-4o states it is correct.
- True Negative (TN): The number of cases where both the human and the GPT-4o agree that the response is incorrect.
- Accuracy (Agreement): The number of cases where both the human and the GPT-4o agree. Accuracy (Agreement rate) = (TP + TN) / (TP + TN + FP + FN)

First, we calculated false negative and false positive cases between human and GPT-4o evaluations for each step, attribute, and question answer. Then we computed the accuracy also called the agreement rate. The results of each step regarding the attributes are provided in Tables 12, 13, 14, and 15. The "Question" in tables represents the question answer merged with three properties (number + subject + action).

The results show the agreement rate between human evaluation and GPT-4o was 91% to describe images, 94% to identify relationships, 92% to apply relationships, and 91% to generate images. Although the attribute agreement rate is the lowest at 74%, the precision to answer questions, which require integration properties, exceeds 91%.

Table 12: Step 1 - Describing Images

| Attribute | FP | FN | TP + TN | Accuracy |
|---|---|---|---|---|
| Number | 1 | 7 | 142 | 95% |
| Subject | 5 | 5 | 140 | 93% |
| Action | 5 | 14 | 131 | 87% |
| Question | 6 | 7 | 137 | **91**% |

Table 13: Step 2 - Identifying Relations

| Attribute | FP | FN | TP + TN | Accuracy |
|---|---|---|---|---|
| Number | 4 | 5 | 39 | 78% |
| Subject | 1 | 12 | 37 | 74% |
| Action | 2 | 8 | 40 | 80% |
| Question | 1 | 2 | 47 | **94**% |

Table 14: Step 3 - Applying Relationships

| Attribute | FP | FN | TP + TN | Accuracy |
|---|---|---|---|---|
| Number | 3 | 8 | 39 | 78% |
| Subject | 4 | 7 | 39 | 78% |
| Action | 8 | 4 | 38 | 76% |
| Question | 2 | 2 | 46 | **92**% |

Table 15: Step 4 - Generating Images

| Attribute | FP | FN | TP + TN | Accuracy |
|---|---|---|---|---|
| Number | 5 | 2 | 23 | 77% |
| Subject | 1 | 4 | 25 | 83% |
| Action | 2 | 4 | 24 | 80% |
| Question | 1 | 2 | 27 | **90**% |

## B.6 ANYRES STRATEGY WITH IMAGE COLLAGES

To investigate whether the AnyRes strategy would improve the performance of collaged images, we experimented with the LLaVA-OneVision (Li et al., 2024) on VOILA-ND utilizing both image collage and multiple image formats. We tested the model only in the first stage where the images were processed. The model utilizing image collage achieves an accuracy of 53%, while the model using multiple images performs slightly better, with 57% accuracy in describing images. Table 16 summarizes the results, showing that the AnyRes approach improves the image resolution and closes the performance gap between the process of image collage and multiple images.

Table 16: Performance of LLaVA-OneVision on different input types

| Method | Input Type | Number | Subject | Action | Total |
|---|---|---|---|---|---|
| LLaVA-OneVision | Image Collage | 63.8% | **94.5**% | **84.3**% | 53% |
| LLaVA-OneVision | Three Sequential Images | **67.9**% | 73.7% | 67.5% | **57.5**% |

## B.7 IMPACT OF SUB-PROMPTS ON IDENTIFYING RELATIONSHIPS

We experimented with GPT-4o on the VOILA-WD dataset to discover the impact of employing sub-prompts for determining property relationships between first and second images. For a fair

comparison, we froze the first-step answers and requested the model to find whether the number/subject/action changed or remained the same from the first image to the second image. After merging the results from three sub-tasks, we achieved a similar accuracy of 42% with a single prompt experiment. The accuracy of properties is also similar with 94% for numbers, 79% for subjects, and 56% for action. The relationship identification performance of GPT-4o with single and three sub-prompts, provided in Table 17 demonstrates that GPT-4o's ability to identify relationships is not affected by the number of properties asked in the prompt.

Table 17: Performance of GPT-4o on identifying relationships with different prompting approaches

| Model | Approach | Number | Subject | Action | Total |
|---|---|---|---|---|---|
| GPT-4o | Single Prompt | **94.3**% | 78.5% | 55.9% | **42.8**% |
| GPT-4o | Three Sub-prompts | 94.1% | **79.3**% | **56.3**% | 42.3% |

### B.8 CoT vs L2M For Applying Relations

To evaluate the effectiveness of CoT in the "Applying Relationship" step, we conducted an experiment utilizing GPT-4o on the VOILA-WD dataset. For a fair comparison of implementing CoT and L2M for step 3, we froze the first and second-step answers and provided two textual examples and their rationales. We requested from model to find the properties of the fourth image by providing previous sub-task answers. The result of the study provided in Table 18 shows that the L2M approach with detailed instructions performs slightly better than the CoT approach for this task.

Table 18: Performance of GPT-4o with different approaches

| Model | Approach | Inputs | Number | Subject | Action | Total |
|---|---|---|---|---|---|---|
| GPT-4o | CoT | Two Examples | **48.8**% | 33.1% | 29% | 5.96% |
| GPT-4o | L2M | Instructions | 33.8% | **36.5**% | **35.0**% | **6.44**% |

### B.9 Human vs Models Using Examples and Options

We conducted an ablation study using GPT-4o on the VOILA-WD dataset to evaluate the performance gap between human and model responses in the same conditions where two example questions and property options are provided. As the human evaluation was performed in one step, we set up the experiment for the direct answering approach. In addition to the example images and lists, we also provided detailed rationales to support the solutions. The experimental results provided in Table 19 demonstrate that offering examples and option lists impacts the model's performance similar to the L2M approach employing step-by-step instructions. Although the model selects each property answer from the given list and analyzes example images and logic, its performance is significantly below the human performance, with a notable 65% accuracy gap.

Table 19: Performance comparison between human evaluation and models

| Method | Approach | Inputs | Number | Subject | Action | Total |
|---|---|---|---|---|---|---|
| Human | Direct Answering | Two Examples and List | **93.6**% | **82.3**% | **91.4**% | **71.36**% |
| GPT-4o | Direct Answering | Two Examples and List | 46.6% | 25.28% | 64.4% | 6.8% |
| GPT-4o | L2M | Instructions | 33.8% | 36.5% | 35.0% | 6.44% |

### B.10 Failure Case

An example of how visual analogy questions evaluate the MLLMs step-by-step is provided in Figure 14. In the example, we employ GPT-4o using the L2M approach with a sequence of three images.

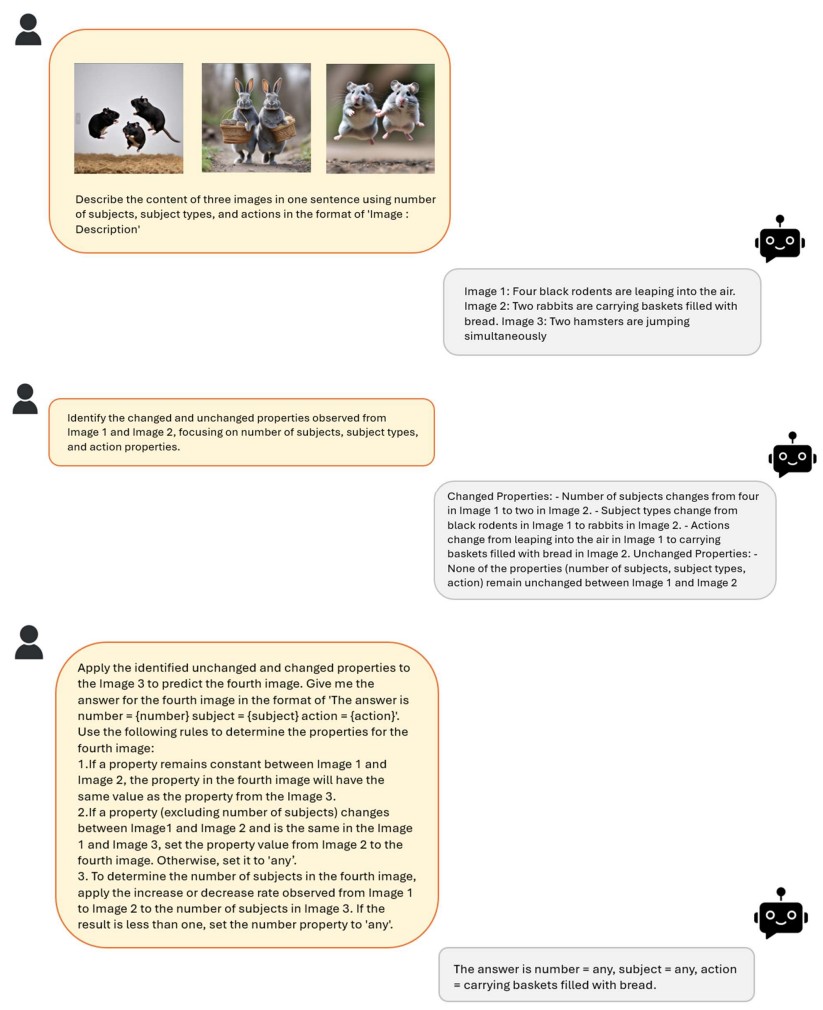

Figure 14: Failure L2M case of GPT-4o on `VOILA-WD`

## B.11 SUCCESS CASE

An example of how GPT-4o successfully reaches the solution is provided in Figure 15. To show all stages of the task, including image generation, we utilize GPT-4o in the example.

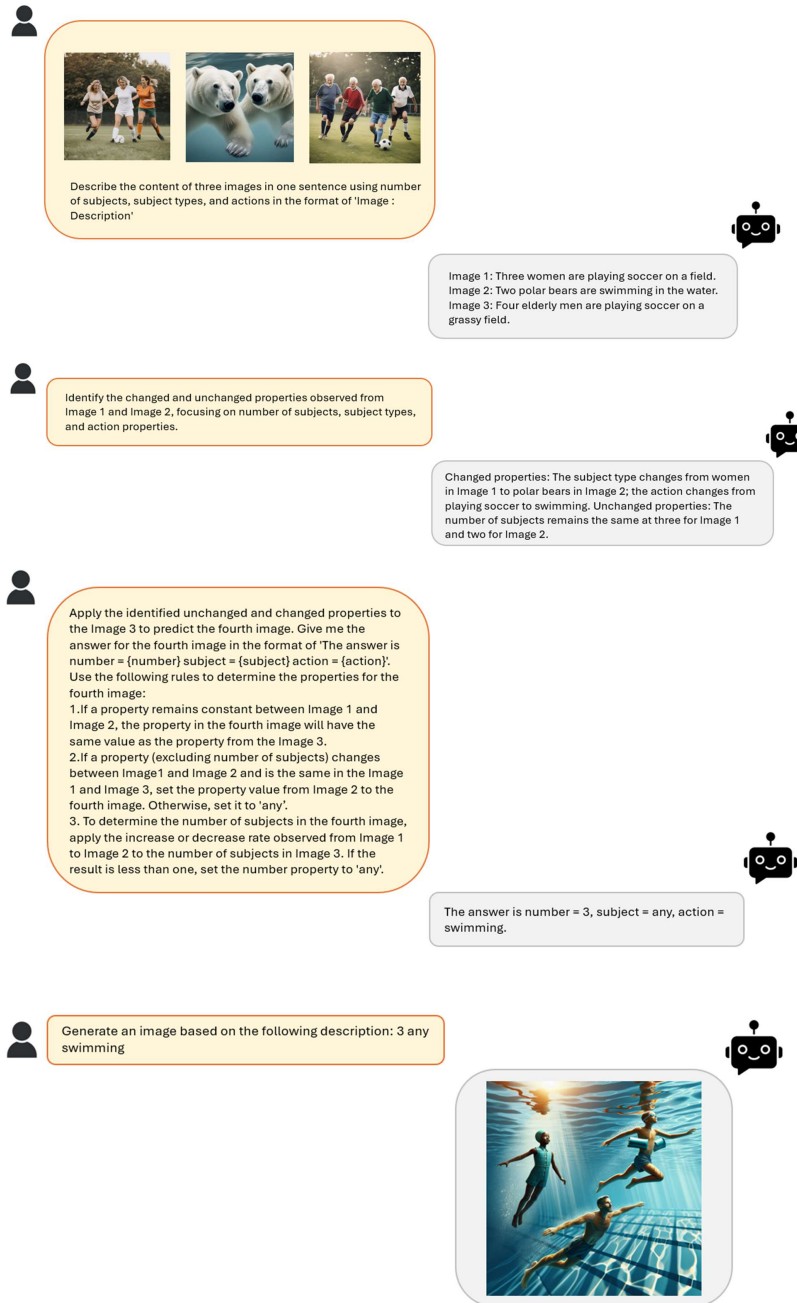

Figure 15: Success L2M case of GPT-4o on `VOILA-WD`

## B.12 EVALUATION PROMPTS STEP BY STEP

After receiving the answers from MLLMs for using the L2M method, we execute the evaluation pipeline which consists of multiple stages. Figures 16, 17, 18, and 19 show the evaluation of the models' outputs step by step.

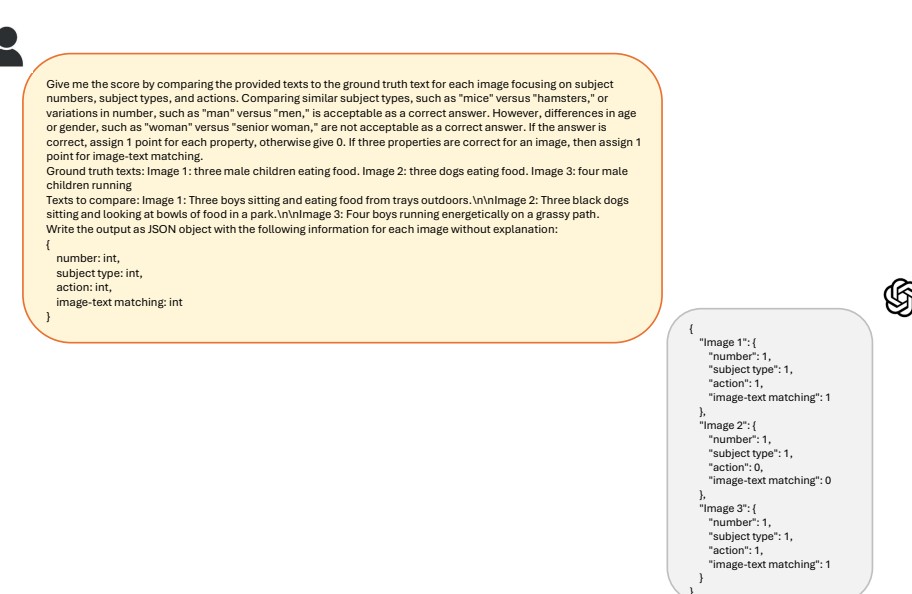

Give me the score by comparing the provided texts to the ground truth text for each image focusing on subject numbers, subject types, and actions. Comparing similar subject types, such as "mice" versus "hamsters," or variations in number, such as "man" versus "men," is acceptable as a correct answer. However, differences in age or gender, such as "woman" versus "senior woman," are not acceptable as a correct answer. If the answer is correct, assign 1 point for each property, otherwise give 0. If three properties are correct for an image, then assign 1 point for image-text matching.
Ground truth texts: Image 1: three male children eating food. Image 2: three dogs eating food. Image 3: four male children running
Texts to compare: Image 1: Three boys sitting and eating food from trays outdoors.\n\nImage 2: Three black dogs sitting and looking at bowls of food in a park.\n\nImage 3: Four boys running energetically on a grassy path.
Write the output as JSON object with the following information for each image without explanation:
{
    number: int,
    subject type: int,
    action: int,
    image-text matching: int
}

```
{
    "Image 1": {
        "number": 1,
        "subject type": 1,
        "action": 1,
        "image-text matching": 1
    },
    "Image 2": {
        "number": 1,
        "subject type": 1,
        "action": 0,
        "image-text matching": 0
    },
    "Image 3": {
        "number": 1,
        "subject type": 1,
        "action": 1,
        "image-text matching": 1
    }
}
```

Figure 16: Step 1 evaluation

Give me the score by comparing the provided texts to the ground truth text to determine if the model correctly identifies the unchanged and changed properties between Image 1 and Image 2. The properties to compare are the number of subjects, the subject type, and the actions. If the answer is correct, assign 1 point for each property, otherwise give 0. If three properties are correct then assign 1 point for identify_changes accuracy.
Ground truth texts: Number remains constant three. Action remains constant eating food. Subject type is changed from male children to dogs.
Texts to compare: From Image 1 to Image 2:\n\n- Changed Properties:\n - Subject Types: Changed from \"boys\" to \"dogs.\"\n - Action: Changed from \"sitting and eating food from trays outdoors\" to \"sitting and looking at bowls of food in a park.\"\n\n- Unchanged Properties:\n - Number of Subjects: Remains three.
Write the output as JSON object with the following information for each image without explanation:
{
    number: int,
    subject type: int,
    action: int,
    identify_changes: int
}

```
{
    "number": 1,
    "subject type": 1,
    "action": 0,
    "identify_changes": 0
}
```

Figure 17: Step 2 evaluation

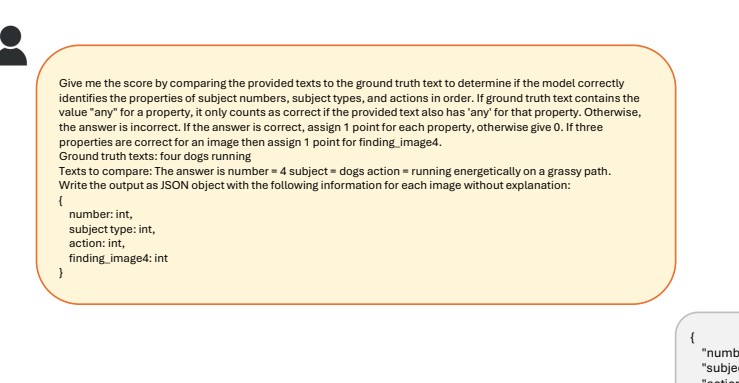

Figure 18: Step 3 evaluation

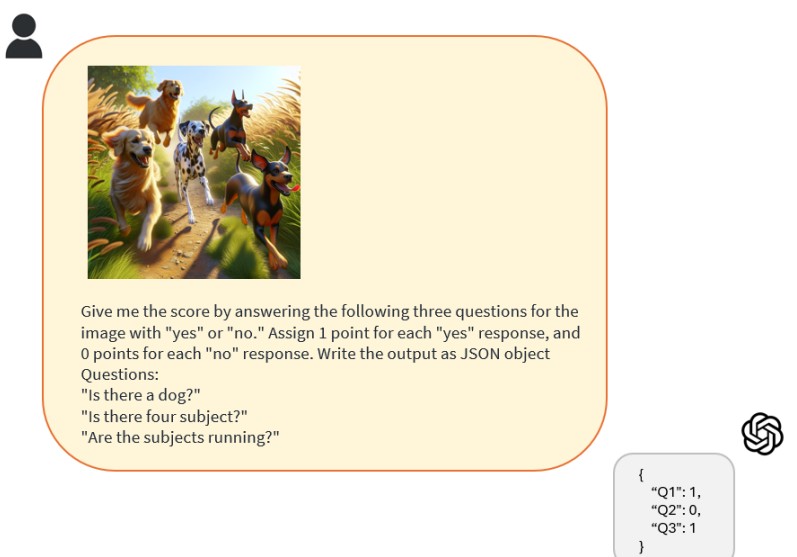

Figure 19: Step 4 evaluation

## C PROMPTS

### C.1 PROMPT SELECTING

To find the best-performing parameters, we tested various text prompts on MLLMs on VOILA-ND and VOILA-WD utilizing 0-shot and least-to-most prompting. Since VOILA-WD includes distractions, we modify the text prompts for VOILA-ND by excluding the explanation of distraction rules. Figure 20 displays diverse zero-shot prompts we tried before selecting the final instructions. For L2M prompting, we need multiple text instructions for each step. Various prompts are tested for each sub-problem in a multiple-step reasoning process, see Figures 20, 21, 22, and 23.

> You are an AI visual assistant who can answer the visual analogy questions. To answer such questions:
> 1- You should first understand the visual content of three images, and then find the relationships between the first and second images by looking at changes in numbers, subject types, and actions. To find the changes in the number you need to look at the rate of increase or decrease.
> 2- Then apply same relationship to the third image to determine the fourth image. Give me the answer to what the fourth image is in the format of "The answer is number = {number} subject = {subject} action = {action}"
> If the subject number in the fourth image is less than one, the number in the fourth image is 'any'. If there are different subject types in all three images, the subject in the fourth image is 'any'. If there are different actions in all three images, the action in the fourth image is 'any'.

> You are an AI visual assistant who can analyze visual analogy questions with three images and find the fourth image based on the relationships observed in the first three. Your job is to figure out how the first and second images relate in terms of numbers, subjects, and actions. The relation of numbers depends on the rate of increase or decrease. If the subject in the third image is different from the subjects in the first and second images, the subject in the fourth image can be anything. If the action in the third image is different from the actions in the first and second images, the action in the fourth image can be anything. What is the fourth image? "The answer is number = {number} subject = {subject} action = {action}"

> Complete the visual analogy by comparing at subject type, number, and action properties: A is to B as C is to? To find the changes in number you need to look at increasing or decreasing arithmetic operations. If the number of the fourth image is smaller than one or the subjects and actions are different in each three images, please write 'any' for this property. Find the fourth image in the format of "The answer is number = {number} subject = {subject} action = {action}"

> Complete the visual analogy by comparing at subject type, number and action properties: A is to B as C is to? Answer in a format of "The answer is: number = {number} subject = {subject} action = {action}"

> Complete the visual analogy, A is to B as C is to D, by comparing at object type, number and action properties. Number can be find in decreasing or increasing order. What is the number of object, object type and action in fourth image?

Figure 20: Zero-shot prompt selection.

> Prompt 1: Describe the content of the images in format of number = {number} subject= {subject} action = {action}

> Prompt 1: Describe the content of three images in one sentence using numbers, subjects, and actions in the format of "Image: Description"

> Prompt 1: The image consists of three images. Describe the content of three images in one sentence using numbers, subjects, and actions.

Figure 21: First prompt selection of L2M process

> Prompt 2: Identify the relationship between the first and second images by looking at changed or stable pattern in numbers, subject types, and actions.

> Prompt 2: Identify the changed and unchanged properties observed between the first and second images, focusing on number of subjects, subject types, and action properties. If the property in the first image is different from the second image, use the format of "property is changed from X to Y" else use "property remains constant X.

> Prompt 2: Identify the changed and unchanged properties observed between the first and second images, focusing on number of subjects, subject types, and action properties. For the number of subjects, consider the change in either increase or decrease

Figure 22: Second prompt selection of L2M process.

Prompt 3: Apply the unchanged and changed number, subject, and action properties to the third image to find the fourth image. If a property is constant, it remains the same in the fourth image. If the property is changed, and it is the same in the first image, then the property in the fourth image will be the same in the second image. Otherwise, it can be 'any'. What is the fourth image? Write the answer in the format of "The fourth image is number = {number} subject = {subject} action = {action}"

Prompt 3: Apply the unchanged and changed property rules to the third image to predict the fourth image. What is the fourth image? "The answer is number = {number} subject = {subject} action = {action}".If property except number is changed and different in all three images, the property in the fourth image can be 'any'.

Prompt 3: Apply the unchanged and changed number, subject, and action properties to the third image to find the fourth image. If a subject is constant, it remains the same in the fourth image. If the subject is changed, and it is the same in the first image, then the subject in the fourth image will be the same in the second image. Otherwise, it can be 'any'. If an action is constant, it remains the same in the fourth image. If the action is changed, and it is the same in the first image, then the action in the fourth image will be the same in the second image. Otherwise, it can be 'any'. What is the fourth image? Write the answer in the format of "The fourth image is number = {number} subject = {subject} action = {action}".

Prompt 3: Apply rules to predict number of subject, subject and action in fourth image:
1.If a property is the same in the first and second images, set the property value from the third image to the fourth image.
2.If a property (excluding number of subjects) changes between the first and second images and is the same in the first and third images, set the property value from the second image to the fourth image. Otherwise, set it to 'any'.
3.Apply the increase or decrease rate in the number of subjects from the first to the second image to the third image. If the result is less than one, set the number to 'any'.
Format your prediction as:'The answer is number = {number} subject = {subject} action = {action}'

Prompt 3: Study the content of the three images in series, identify the changed and unchanged properties between the first image and the second image. If a property remains constant between first and second images, the corresponding property in the fourth image will have the same value as that of the third image. If a property (excluding the number of subjects) changes between the first and second images and is the same in the first and third images, the value of the respective property in the fourth image will be the same as that of the second image.
Predict the description of the hidden fourth image in format of 'The answer is number = {number} subject = {subject} action = {action}'

Figure 23: Third prompt selection of L2M process.

# D  HUMAN EXPERIMENT

## D.1  MTURK INSTRUCTIONS

The human study was conducted by using the Amazon Mechanical Turk platform. Participants solve 440 various analogy questions in total. The instructions and example questions of the study were provided in Figure 24.

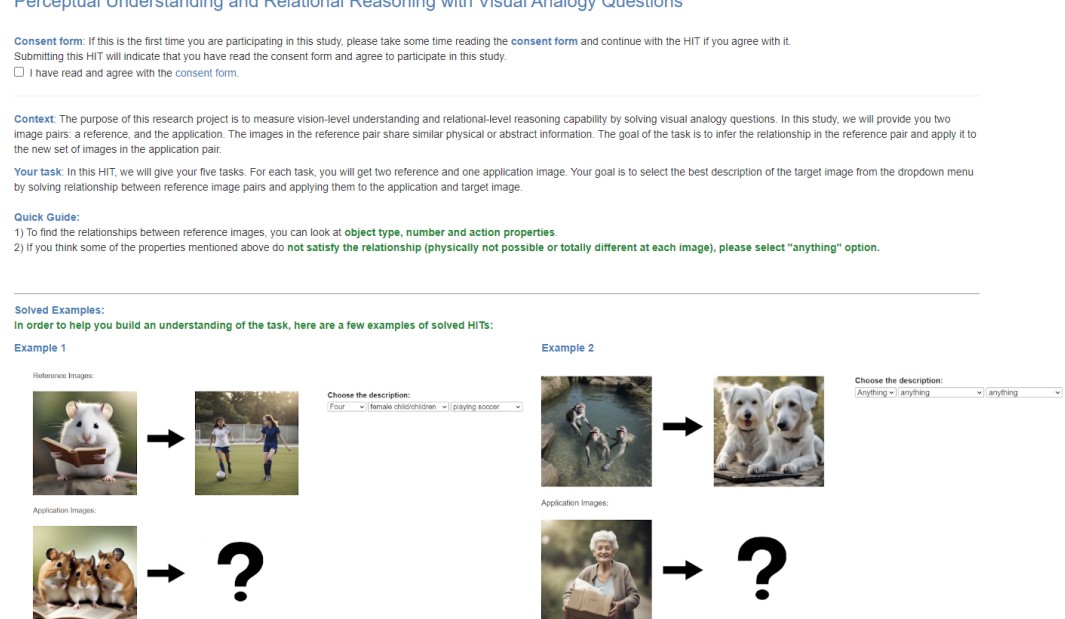

Figure 24: Human experiment of MTurk platform.

