# OpenReview forum: "VOILA: Evaluation of MLLMs For Perceptual Understanding and Analogical Reasoning"
_ICLR.cc/2025/Conference — ICLR 2025 Poster_

### Official Review · Reviewer_pi5r · 2024-10-28

**Soundness:** 2
**Presentation:** 2
**Contribution:** 3
**Rating:** 5
**Confidence:** 4

**Summary:**

The paper introduces VOILA, a benchmark designed to assess multimodal large language models (MLLMs) on tasks requiring visual perception and abstract relational reasoning. VOILA primarily focuses on visual analogical reasoning, where models must generate an image that completes an analogy given two pairs of images. The benchmark includes two subsets, VOILA-WD (with distractions) and VOILA-ND (without distractions), testing MLLMs on varying levels of complexity. Contributions:
1. A dataset creation pipeline allowing large-scale generation of visual analogy questions, and a large-scale, open-ended benchmark to evaluate MLLMs’ high-level visual reasoning capabilities.
2. Comprehensive evaluation of state-of-the-art MLLMs, highlighting performance gaps in relational reasoning compared to human baselines.
3. Ablation studies reveal model limitations in handling complex visual relationships, suggesting areas for improvement in future MLLM development.

**Strengths:**

1. The proposed problem is a novel multimodal reasoning problem, which is well-defined and advances abstract relational reasoning in MLLMs​.
2. The benchmark design is comprehensive, encompassing various rule configurations and levels of task difficulty.
3. The results reveal the weaknesses of current multimodal LLMs.

**Weaknesses:**

1. The paper does not address why the proposed problem is an essential and meaningful problem to solve. In what applications/scenarios is the capability essential?
2. The comparison between the model and human performance may not be fair: (1) Humans are given two examples. (2) It seems the task for humans is to choose properties from available options, while MLLMs need to predict those properties.
3. Table 3 takes up a lot of space and has a lot of numbers, but it is barely mentioned or explained in the text.

**Questions:**

1. The prompt4 for generating images seems unnecessary for evaluating the "reasoning" capabilities of MLLMs. Also, the reviewer is curious how the authors use GPT4o to generate images since it is not clear whether they call another image generation model to generate images in the web interface. And it seems gpt4o API does not support image output.
2. Why Table 4 and Table 5 only use the selected models for the ablation study?

---

> ### Author Response · Authors · 2024-11-19
> **Response to Reviewer pi5r (part 1)**
>
> Thank you for your thoughtful review of our paper, VOILA. We are pleased that you recognized our work as **novel, well-defined, and comprehensive** with various properties, rule configurations, and difficulty levels designed to **reveal the shortcomings of state-of-the-art MLLMs through detailed and comprehensive evaluations**. We are delighted that you recognize the significance of VOILA in **making advancements to abstract relational reasoning in MLLMs.** Below, we provide detailed responses to your specific inquiries:
>
> **“Why the proposed problem is an essential and meaningful problem to solve. In what applications or scenarios is the capability essential?”**
>
> Analogical reasoning, commonly used in IQ tests, is a critical component for assessing cognitive abilities, encompassing **problem-solving, decision-making, classification, and learning** [7]. It involves the process of mapping abstract relationships between concepts and applying these relational patterns, which is fundamental for higher-level cognitive assessments. This reasoning is crucial to improve the **generalization ability** in which the model implements the identified pattern to a new set of data. Transferring knowledge from identified relationships allows models to make human-like decisions with complex abstract reasoning abilities across various contexts.
>
> Analogical reasoning is essential when **transferring the previous relational learnings from one concept to another.** It can be applied in many application areas from medical diagnostics to creative industries. One example is autonomous vehicles that can use analogical reasoning for decision-making during navigation by recognizing the relationship between signs, intersections, or traffic flow patterns.
>
> We will add more information to the paper to explain the significance of analogical reasoning.
>
> **“The comparison between the model and human performance may not be fair: (1) Humans are given two examples (2) It seems the task for humans is to choose properties from available options, while MLLMs need to predict those properties.”**
>
> We believe that the comparison between human and MLLM performance is fair. Human performances are conducted by providing two examples to explain the Voila-ND and Voila-WD dataset discrimination. However, no direct instructions were provided to human performers as they were given to the models with prompts. Instead of image examples, we implemented text-based instructions for models by following the L2M approach.  Since our ablation study 6.2 shows that MLLMs are better at processing textual inputs than visual data, the experimental setup benefits them to understand the task better than visual examples.
>
> Providing options may appear to influence human performance for the first step, however, **to detect the distraction property relations (any), participants need more than just selection; they must perform a logical reasoning process to understand and apply the relevant and irrelevant relations** provided in the instruction to the models. To facilitate the assessment of the human participants’ answers, we provided 14 actions, 15 subjects, and 5 numbers including the ‘any’ option. This configuration results in an **accuracy of approximately 0.095% for random selection** which is significantly lower than the **human performance of 70%**. As participants complete the visual analogy questions in a single step—selecting the best answer from the available options without intermediate stages—the gap between random selection and human performance indicates that the provided options do not offer a meaningful advantage.
>
> Table 3, including the detailed model performance on VOILA-WD, shows that most of the models, excluding CogVLM2 and LLaVa, correctly recognize three properties with above 80% accuracy at the first step. This demonstrates that models are capable of describing the images by predicting the properties. Providing options might increase the accuracy of the first step, however, the performance of the models drops significantly after identifying relations and application relationships step, as seen in Table 3 with comprehensive property analysis at each step. This confirms that **establishing logical relations, discovering irrelevant properties, and applying relationships require more than simply recognizing the properties; it demands high-level abstract reasoning capabilities.** Another study in section 6.1 supports these findings. Although models are provided with ground truths at each step, their performance in the third step only achieves **17% accuracy**.  As offering examples and options does not compromise the core logic of the task, we believe our human vs model comparison is fair.
>
> ---
> [7] Goswami, U. (1992). Analogical Reasoning in Children. Psychology Press. https://doi.org/10.4324/9781315804729

---

> > ### Author Response · Authors · 2024-11-19
> > **Response to Reviewer pi5r (part 2)**
> >
> > **“Table 3 takes up a lot of space and has a lot of numbers, but it is barely mentioned or explained in the text.”**
> >
> > Thank you for pointing this out. Table 3 includes a comprehensive and detailed property analysis of models at each stage on VOILA-WD. Section 5.2 explains the key findings presented in Table 3 with Figure 5, which visually represents a subset of data in Table 3. We will add more key insights for Table 3 to the paper and make sure it is well explained in the final draft.
> >
> > **“The prompt 4 for generating images seems unnecessary for evaluating the "reasoning" capabilities of MLLMs.”**
> >
> > While image generation in standard visual analogies might have a limited impact on evaluating the reasoning ability of the MLLMs because the main focus of the task is relational mapping and its application. However, in VOILA, image generation plays a more active role in reasoning evaluation with the applied distraction rule. After the third step, the correct text prompts can contain “any” keywords for indicating irrelevant property relations. To generate semantically aligned and consistent images, the model must understand and apply the content from these text prompts. It evaluates how effectively the model synthesizes text data and translates it into the visual domain without losing the semantic information of the text. Even if the models capable of generating images reach the correct text-based outcome after the third step, without generating accurate images the answer is wrong. The experiment results in section 5.1 and Appendix D.2 show the impact of the image generation step on the models’ performance, highlighting a 10% decrease in accuracy.
> >
> > **“How do the authors use GPT4o to generate images?”**
> >
> > We used DALL-E 3 which is integrated with the GPT models [8]
> >
> > **“Why do Table 4 and Table 5 only use the selected models for the ablation study?”**
> >
> > As only QWEN-VL2 and GPT-4o models accept both image collage and sequential image formats, the effect of the input formats was tested on two models in Table 5.
> > After models tested on the direct answering approach, performed with very low accuracy, we switched to the L2M method to improve their performance. L2M helps models to generate more accurate results and because of time and resource constraints, we did not prefer to conduct more studies using direct answering. We believe the results in Table 4 are sufficient and consistent to demonstrate the improvement of using the multiple-step approach in complex reasoning tasks.
> >
> > We trust that our response addresses your concerns comprehensively and encourage you to reevaluate our submission positively. We look forward to further discussion.
> >
> > ---
> > [8] https://openai.com/index/dall-e-3/

---

> > ### Comment · Reviewer_pi5r · 2024-11-25
> > **Re: part 1**
> >
> > Regarding Weakness 2, the response addresses it to some level. However, offering examples (in-context learning) has been shown to be very helpful in various tasks in LLM / MLLMs without compromising the core logic of those tasks. Therefore, it is still quite possible that providing examples (or text only logic reasoning examples) could improve the performance.
> >
> > Moreover, a basic prompting method that has been proven useful, chain-of-thought reasoning, could be tried on the MLLMs. For instance, in the "Applying Relationship" step, chain-of-thought can be applied instead of letting the models directly output the result. This is a very intuitive and easy-to-implement step, without which the abilities of MLLMs may not be well-presented (also, humans are allowed to perform these reasoning steps while answering the question). The above two points make the abilities of MLLMs and the performance gap of MLLMs from humans may not be well-evaluated.
> >
> > Other responses from the review address some of the concerns and make the motivation more clear. I've raised the score.

---

> > > ### Author Response · Authors · 2024-11-28
> > >
> > > Thank you for considering our previous responses and for raising your score. We appreciate your feedback and would like to address the remaining concerns you have. Below, we provide additional clarifications and ablation studies on the comparison between human and model performance and CoT use for the “Applying Relationship” step:
> > >
> > > **Comparing human performance with model results using examples and options**
> > >
> > > We conducted a new ablation study utilizing GPT-4o on the VOILA-WD dataset to address your concern about evaluating the performance gap between human and model answers. The study aims to compare human performance with model results in the same conditions where two example questions and property options are provided. Since the human evaluation was performed in one step, we set up the experiment for the direct answering approach. In addition to the example images and lists, we also provided detailed rationales to support the solutions. The experimental results demonstrate that **providing examples and option lists impacts the model’s performance similarly to the least-to-most (L2M) approach with step-by-step instructions**. It confirms that even **without examples or pre-defined option lists, the capabilities of MLLMs are effectively presented when guided by multi-step instructions**. Although the model selects each property answer from the given list and analyzes example images and logic, its performance is significantly below the human performance, with a **notable 65% accuracy gap**. This difference highlights the limitations of models in human-level analogical reasoning and validates our prior findings on the gap between human and model performance. The details of human and model performances are provided in the table below.
> > >
> > > Method       |    Approach   |    Inputs       | Number       | Subject       |     Action    |    Total     |
> > > ------------------|-----------------|-----------------|-------------------|------------------|------------------|--------------|
> > >  **Human** | Direct Answering | Two Examples and List | 93.6% | 82.3% | 91.4% | **71.36%**   |
> > >  **GPT-4o** | Direct Answering |Two Examples and List|  46.6% | 25.28% | 64.4%|**6.8%** |
> > > **GPT-4o**   | L2M		| Instructions	| 33.8% | 36.5% | 35.0	%    | **6.44%** 	|
> > >
> > > **Implementing CoT for the “Applying Relations” step.**
> > >
> > > Because one of the concerns is applying a different approach, CoT, for the reasoning process, we conducted a new experiment with GPT-4o on the VOILA-WD dataset to test the effectiveness of CoT in the “Applying Relationship” step. In our study, we utilized the L2M (least to most) prompting method, which adopts a **similar multi-phase reasoning structure as CoT but without rationales**. Instead, we provided detailed instructions and broke down the task, using the solution to each sub-problem as input for the next sub-task.
> > >
> > > For a fair comparison of implementing CoT and L2M for step 3, we froze the first and second-step answers and provided two textual examples and their rationales. We requested from model to find the properties of the fourth image by providing previous sub-task answers. The result of the study provided in the table below shows that the **L2M approach performs slightly better than the CoT** approach for this task. This study ensures that the **reasoning abilities of the MLLMs are well-presented and evaluated** in the paper. This outcome also aligns with the findings of the study [9] demonstrating that L2M obtains better capability to generalize challenging problems than CoT.
> > >
> > > Method       |    Approach   |    Inputs       | Number       | Subject       |     Action    |    Total     |
> > > ------------------|-----------------|-----------------|-------------------|------------------|------------------|--------------|
> > >  GPT-4o | **CoT** | Two Examples | 48.8% | 33.1% | 29% | **5.96%**   |
> > >  GPT-4o | **L2M** | Instructions|  33.8% | 36.5% | 35.0%    | **6.44%** |
> > >
> > > These two experiments confirm the performance gap between MLLMs and humans while demonstrating the effectiveness of evaluating MLLMs using the L2M approach. **Offering examples, rationales and option lists has an equivalent impact on VOILA as employing the multi-step process with instructions**. The implementation of both L2M and CoT highlights the limitations of current MLLMs in effectively performing the analogical reasoning process.
> > >
> > > We believe our response and new experiments address your remaining concerns comprehensively and encourage you to reevaluate our submission positively. We look forward to further discussion.

---

> > > > ### Author Response · Authors · 2024-12-01
> > > > **Follow-Up on Discussion Response**
> > > >
> > > > We sincerely appreciate your invaluable feedback on our paper. To address your remaining concerns, we have provided detailed explanations and conducted two additional experiments to support our responses. As the discussion is nearing its conclusion, we wanted to kindly confirm if our responses have adequately addressed all your concerns or if any remaining points require further clarification.

---

> > > > > ### Comment · Reviewer_pi5r · 2024-12-01
> > > > >
> > > > > Thanks for the extra experiments provided. The experiment results make the conclusion more solid.
> > > > >
> > > > > However, I still have concerns about the image generation step. Firstly, GPT4o and DALL-E 3 are two different models, the errors on DALL-E 3 has nothing related to GPT4o. However, the evaluation target for this paper is MLLMs like GPT4o, not image generation models like DALL-E 3. This makes the image generation result of GPT4o not very relevant to the purpose of the paper. Secondly, the image generation results of emu2 in the paper are missing. Third, there is not much error analysis in this step about how and why the performance dropped.

---

> > > > > > ### Author Response · Authors · 2024-12-03
> > > > > > **Response to Reviewer pi5r**
> > > > > >
> > > > > > The paper focuses on completing the visual analogy question either textual or image-based outputs.  For the image generation step, DALL-E 3 was integrated with GPT-4o due to its compatibility. In addition to DALL-E 3, we evaluated VOILA on four other image-generation MLLMs: SEED-LLaMa, EMU-2, Chameleon, and Gemini. While Chameleon and Gemini were unable to perform the task successfully, we reported results for the three remaining models. As these models are MLLMs capable of generating models, they align closely with the objectives of this paper.
> > > > > >
> > > > > > For the results of EMU-2, please refer to the response for Reviewer qyEy (part 1).
> > > > > >
> > > > > > EMU-2 lacks the ability to accurately depict the number of subjects and, especially, the actions in an image. While SEED-LLaMA occasionally generates the correct number of subjects, its depiction of both subjects and actions is limited. Similarly, DALL-E 3 faces challenges in accurately illustrating the actions in the image. For a detailed analysis, please refer to Section D.2 and Tables 3, 10, and 11.
> > > > > >
> > > > > > Thank you once again for your valuable contribution to the project.

---

> ### Comment · Reviewer_pi5r · 2024-11-25
> **Re: part 2**
>
> Based on the above answers, it is still not clear why the evaluation on the image generation step is helpful in evaluating the MLLMs, given you already have the score for "Applying Relationship."(what ability does the image generation score can reflect and the score of "Applying Relationship." can not?) Because in my understanding, the generated image is based on the output of the "Applying Relationship" step (Please correct if I was wrong). Besides, using the external image generation model DALL-E 3 and using GPT4o for evaluation both introduce noise from the final score of image generation to the MLLM's ability -- DALL-E 3 can make errors in image generation given the correct text input, GPT4o can also make mistakes during evaluation.

---

> ### Author Response · Authors · 2024-11-25
> **Re: Re: part 2**
>
> Thank you for considering our previous responses and for raising your score. We would like to address the remaining concerns you have. Below, we provide additional clarifications on the reasoning ability of the image generation step and evaluation performance of GPT-4o:
>
> **Why the evaluation on the image generation step is helpful in evaluating the MLLMs, given you already have the score for "Applying Relationship." DALL-E 3 can make errors in image generation given the correct text input.**
>
> The image generation errors made by DALL-E 3, EMU-2, and SEED-LLAMA, despite correct text inputs, highlight their limitations in reasoning and synthesis processes. Although these models can generate high-quality images, they struggle to render abstract and complex relationships because they rely on text-based embeddings rather than contextual understanding.
>
> In the visual analogy task, models, capable of generating visual outputs, are required to understand semantically complex relational contexts between properties and visualize them accurately. When utilizing the “any” keyword for irrelevant relations, the models are expected to map abstract concepts and their text-based relations onto visual representation. The challenges models face when integrating multiple concepts and relations in the visual domain reveal their limitations in generalization.
>
> VOILA evaluates the complicated relational logic in both text-based and vision-based forms. The ability to translate textual relationships into visual content measures the models’ performance in connecting semantic interpretation with visual reasoning. Evaluating the image generation step provides insights into the models’ shortcomings in understanding the context and effectively generalizing within the visual domain.
>
> **What ability does the image generation score can reflect and the score of "Applying Relationship." can not?**
>
> Although applying relationships is the most critical and challenging step in VOILA, the task evaluates diverse abilities at each stage. In the first step, the models are evaluated for perceptual and semantic context reasoning via image description. The second step focuses on measuring the models’ relational and comparative reasoning by identifying changed and unchanged properties between images. The third step tests the transfer learning and generalization capability of the models with relational mapping. The last step, image generation, assesses the models’ contextual understanding and visual representation mapping with multimodal alignment.
>
> Although models accept the output of the "Applying Relationship" step as the input prompt in the last step, the abilities measured during the image generation process are distinct from the earlier step. While in the third step, models transfer relations to a new set of data, in the image generation step, models convert textual information to a visual representation by working across multiple modalities.
>
> **GPT4o can also make mistakes during evaluation.**
>
> We have already conducted a comprehensive ablation study to evaluate the accuracy of GPT-4o in each step, including image generation. This study shows that GPT-4o has a 10% error rate when evaluating generated images against the ground truths. For a more detailed explanation and analysis of GPT-4o’s performance, along with calculations, please refer to the response for Reviewer YkdC (part 2). This study ensures that GPT-4o’s evaluation capabilities are sufficiently robust for the comprehensive assessment process, despite its’ challenges of applying identified relationships.
>
> We believe our response addresses your remaining concerns comprehensively and encourage you to reevaluate our submission. We look forward to further discussion.

---

### Official Review · Reviewer_qyEy · 2024-11-03

**Soundness:** 3
**Presentation:** 2
**Contribution:** 2
**Rating:** 6
**Confidence:** 4

**Summary:**

This paper introduces a new benchmark for evaluating multi-modal large language models’ perceptual understanding and abstract relational reasoning through analogy completion. It contributes a dataset creation pipeline, a large evaluation dataset consisting of 10K+ questions, and extensive evaluations of both open-source and proprietary models on this benchmark as well as detailed analyses of their performance on different subtasks.

**Strengths:**

- The idea of using visual analogy completion for evaluating multi-modal large language models is well motivated and somewhat unique
- The visual analogy completion problem is well defined and formulated, with clear definitions of the three properties and four rules.
- The dataset has been manually cleaned to ensure that the generated images match the texts.

**Weaknesses:**

- The experiments can be more complete. Currently, most evaluations focus on the first three steps of the analogy completion task with only a few data points on the image generation stage.
- The paper presentation can be improved. For example, the writing can be improved to highlight the most insightful and interesting takeaways.
- While the visual analogy task is interesting, the reviewer is uncertain about the practical value of this benchmark especially given the abundance of multi-modal benchmarks – for example, why should the community use this benchmark to evaluate models’ relation understanding instead of existing VQA benchmarks that can also test for relation understanding?

**Questions:**

- Did the authors include examples of each subtask in the prompts?
- The authors mention they evaluated Emu-2 on the image generation subtask in 5.1, but Table 3 is missing its numbers?
- In Figure 6, it’s interesting that LlaMa 3.2 is the only model with higher performance on Voila-WD than Voila-ND, while all other models exhibit large drops from Voila-ND to Voila-WD regardless of their absolute accuracies. Can the authors provide insights into why this occurs for LlaMa 3.2?
- Nit: table 4, it’s VOILA-ND in both columns – should one of them be VOILA-WD instead?

Suggestions:
- While the authors motivate the visual analogy task by saying that it tests for higher-order reasoning, they break down the task into four subtasks, each of which is an easier task (despite being still difficult to the model). While the reviewer agrees with the authors that this task decomposition enables fine-grained analyses of models’ weaknesses, it’d also be interesting to see an evaluation of the end-to-end analogy completion task.
- Table 2 can highlight the performance drops from the previous to the next stage
- The authors mentioned that models struggle with the visual analogy task when images are combined in a collage format due to resolution constraints. It’d be interesting to see if this finding holds with high resolution image collages for multi-modal models that support the AnyRes strategy such as Llava-OneVision.

---

> ### Author Response · Authors · 2024-11-20
> **Response to Reviewer qyEy (part 1)**
>
> We are encouraged by your review and appreciate your comprehensive evaluation of our paper. We are pleased that you recognize VOILA as a **well-motivated, unique, well-formulated, well-defined** benchmark, and thoroughly supported by **detailed analyses** based on **manually cleaned data**. We are especially pleased that you acknowledge the VOILA’s **contribution to a dataset creation pipeline.**  Please find our detailed responses to your specific inquiries below:
>
> **“Most evaluations focus on the first three steps of the analogy completion task with only a few data points on the image generation stage.”**
>
> Because of the limited number of MLLMs capable of generating and processing multiple/collaged images and understanding the instructions in literature, we obtain more experimental results on text-based answers than image-based answers. We tested VOILA on 5 image generation MLLMs: GPT-4o, SEED-LLaMa, EMU-2, Chameleon, and Gemini. However, in most cases, Chameleon did not generate the output: “I'm unable to meet that request.” and for the rest of the several cases, it did not follow the instructions. On the other hand, Gemini accepts one image input but rejects working on human images. These restrictions might limit our comprehensive evaluation for the image generation step, however, results on the most preferred models in the field; GPT-4o, SEED-LLaMa, and EMU-2, are sufficient to have an idea of how successful state-of-art models to visualize the provided text data.
>
> **“The writing can be improved to highlight the most insightful and interesting takeaways”**
>
> We will review and make changes to the final draft to highlight the most important outcomes of the experiments.
>
> **“Why should the community use this benchmark to evaluate models’ relation understanding instead of existing VQA benchmarks that can also test for relation understanding?”**
>
> Thank you for your excellent question which reveals the significance of the VOILA. The visual analogy task, VOILA, consists of four distinct but related stages, and identifying the relation is one of them. The critical part of the study is **how models apply the defined relationship to new data which differs VOILA from existing VQA benchmarks.** The ablation study on 6.1 highlights the pivotal role of the relationship application process, as the model provided with ground truth relationship values still achieves only 17% performance. Additionally, VOILA is more systematic, better formulated, and better structured than general VQA benchmarks. Thus, it provides additional critical insights (as we discussed in the draft) about MLLM models.
>
> The process of mapping abstract relationships between concepts and applying these relational patterns is a key element in IQ tests and cognitive assessments, where it helps evaluate high-level reasoning and problem-solving. To assess the cognitive intelligence of the model we need more than VQA benchmarks that focus on image description or identifying relations. VOILA provides the required logic to evaluate the models’ inference capacity in high-order relationships and the knowledge transfer ability of these relationships.
>
> What differentiates VOILA from other analogy benchmarks is implementing **distraction rules which expect models not only to understand the relationship and pattern between images but also to discover the irrelevant changes among properties.** While human performers successfully detect the pattern-breaking features in the visual analogy questions, our experiment in 5.4 shows how current models struggle with revealing unrelated properties.
>
> **“Did the authors include examples of each subtask in the prompts?”**
>
> No, the examples are not given in the prompt. Instead, we use the L2M method and give instructions to generate the answer for each step.
>
> **“Table 3 is missing Emu-2 image generation values”**
>
> Thank you for your detailed review. We will add the accuracy of the image generation step of EMU-2 to Table 3. Our results on both VOILA-WD and VOILA-ND show that EMU-2 lacks the ability to accurately draw the number of subjects and especially actions in the image. The image generation performance of EMU-2 on both VOILA-WD and VOILA-ND is provided in the Table below.
>  Method    |    Dataset          | Number         | Subject         |       Action    |    Total     |
> --------------|----------------------|-------------------|------------------|------------------|--------------|
>  **EMU-2**| VOILA-ND    | 0.43%             | 0.73%            |  0.29%          | **0.11%** |
>  **EMU-2**| VOILA-WD     | 0.18%            | 0.23%           | 0.08%          |  **0.03%**      |

---

> > ### Author Response · Authors · 2024-11-20
> > **Response to Reviewer qyEy (part 2)**
> >
> > **“Can the authors provide insights into why LlaMa 3.2 is the only model with higher performance on Voila-WD than Voila-ND?”**
> >
> > We examined the rule-based performances of LlaMa 3.2 to investigate the accuracy differences between VOILA-WD and VOILA-ND. We recognized that LlaMa 3.2 answers more correct analogy questions on rules 2, 4, 6, 8, 12, and 14 on the VOILA-WD dataset. The questions applied to rules 4, 8, and 12 consist of two distraction rules for distinct properties. Additionally, in five rule configurations, the distraction rule is assigned to the number property. This shows that **LlaMa identifies distractors and applies the distraction rule better than other rules (arithmetic and stable)**, especially in number property which explains why it achieves better results on the VOILA-WD dataset rather than VOILA-ND, unlike other models. We will add this explanation to the paper.
> >
> > **“Table 4, it’s VOILA-ND in both columns – should one of them be VOILA-WD instead?”**
> >
> > Yes, the name of the first column is VOILA-WD. Thank you for your detailed review. We will correct it.
> >
> > **“Evaluation of the end-to-end analogy completion task.”**
> >
> > In Section 5.3, we conducted experiments including end-to-end analogy completion tasks to compare the performances of the models applying L2M and direct answering approaches including a set of instructions in the prompt on both the VOILA-WD and VOILA-ND dataset. Cog-VLM2, GPT-4o, and SEED-LLaMa were tested on both approaches with different prompts. The results are given in Table 4.
> >
> > Also, in the ablation study 6.2, we conducted an end-to-end analogy completion study on the VOILA-ND dataset using a direct answering prompt without explanation of rules: "Complete the visual analogy, A is to B as C is to D, by comparing subject type, number, and action properties. What is the number of subjects, subject type, and action in the fourth image? Answer in a format of ‘The answer is number = {number} subject = {subject} action = {action}’". In that scenario, we tested whether the given instructions helped the model reach the correct answer.
> >
> > **“Table 2 revision for highlighting the performance drops”**
> >
> > Thank you for your suggestion. We believe that adding highlights would enhance the readability of Table 2 and we will revise the table accordingly to improve the visual clarity.
> >
> > **“Llava-OneVision experiment using AnyRes strategy with high-resolution image collages”**
> >
> > Thank you for the suggestion. As requested, to assess the AnyRes strategy on our dataset, we conducted an experiment with the Llava-OneVision model on VOILA-ND utilizing both image collage and multiple image formats. As we investigated whether the AnyRes strategy would improve the performance of collaged images, we tested the model only in the first stage where the images were processed. The model utilizing image collage achieves an accuracy of 53%, while the model using multiple images performs slightly better, with 57% accuracy in describing images. The experiment results show that the **AnyRes approach improves the image resolution and closes the performance gap between the process of image collage and multiple images.** The results are provided in the table below. We will add this analysis to the paper.
> >
> > Method          | Input Type            | Number         | Subject         |       Action    |    Total     |
> > ------------------|-------------------------|-------------------|------------------|------------------|--------------|
> >  **Llava-OneVision** | Image Collage    | 63.8%      |  94.5%        | 84.3%       |  53%        |
> >  **Llava-OneVision** | Three Sequential Images  | 67.9%  |  73.7% | 67.5%  |  **57.5%** |
> >
> > We believe that our response addresses your concerns comprehensively and encourage you to reevaluate our submission positively. We look forward to further discussion.

---

> > > ### Comment · Reviewer_qyEy · 2024-11-26
> > >
> > > The reviewer thanks the authors for their detailed responses and additional results! They addressed most of the concerns.
> > >
> > > The numbers on Llava-OneVision using AnyRes strategy are especially interesting and suggest that the AnyRes strategy can close the gap between image collage and multiple images as expected.
> > >
> > > However, the reviewer has one main concern left: while the importance of analogical reasoning has been discussed, it is still unclear to me if the decomposed version of this task (using the L2M prompting strategy) still tests for analogical reasoning? Or does it test for other atomic abilities such as relationship understanding (which other benchmarks also evaluate)? In my opinion, the direct answer version of this task is closest to testing high-level analogical reasoning, but the concern there is that the numbers are universally extremely low, making this benchmark's practical value uncertain.

---

> > > > ### Author Response · Authors · 2024-11-26
> > > > **Re: Official Comment by Reviewer qyEy**
> > > >
> > > > Thank you for your insightful comments. Please find our detailed responses to your specific inquiries below:
> > > >
> > > > **Does the decomposed version of this task test for analogical reasoning or does it test for other atomic abilities?**
> > > >
> > > > Both direct answering and L2M approaches test the analogical reasoning skills of models with different implementation formats. While the L2M approach decomposes the problem into multi-step reasoning sub-tasks that align with the cognitive processes involved in solving the visual analogy problem, direct answering performs the task in a single step. The structured, step-by-step approach of L2M breaks down the complex problem into manageable parts and simulates human problem-solving strategies for analogical reasoning tasks.
> > > >
> > > > Through the sequential process of each sub-task, VOILA **evaluates the main analogical reasoning ability alongside diverse atomic abilities** such as perceptual understanding, comparative relational reasoning, and transferring relationships. These **atomic abilities are not separate from the analogical reasoning process, on the contrary, they collectively constitute it**. Since they are integral sub-tasks of the analogical reasoning process, their integration is essential to achieve a coherent solution for the problem. While L2M provides a multi-stage reasoning process with a more guided approach, it does not alter the main goal of analogical reasoning. Instead, it enhances the model’s ability to achieve the solution by clarifying the reasoning steps required to address the visual analogy task.
> > > >
> > > > **The practical value of the VOILA**
> > > >
> > > > Visual analogical reasoning is a highly challenging task for the MLLMs, as it requires integrating multiple reasoning skills. The authors believe that both direct answering and L2M methods test the same high-level analogical reasoning but in distinct ways. Direct answering bypasses intermediate stages, resulting in lower performance compared to the L2M approach, and our experiments align with the findings in [9], which demonstrates that a multi-step framework improves the model’s performance on reasoning tasks. However, the low scores in both approaches demonstrate critical gaps and challenges in the analogical reasoning ability of state-of-the-art MLLMs. These challenges of transferring relational knowledge to a new domain must be addressed to solve real-world problems. One example is autonomous vehicles that can use analogical reasoning for decision-making during navigation by recognizing the relationship between signs, intersections, or traffic flow patterns.
> > > >
> > > > VOILA, which dynamically creates over 6.4 million analogy questions, serves as a practical benchmark to improve the models’ visual analogical performance. The models can be trained or fine-tuned on VOILA to develop the required skills and enhance weaknesses such as analogical mapping and relational understanding. With training, the generalization ability of the models across similar reasoning tasks can be improved. Iterative training and evaluation on VOILA provide improvement of MLLMs’ high-level reasoning capabilities and overall effectiveness.
> > > >
> > > > Moreover, utilizing the hierarchical structure of L2M, VOILA reveals new insight into the strengths and weaknesses of MLLMs in solving visual analogy questions and provides more actionable understandings to address specific models’ shortcomings. This comprehensive evaluation goes beyond the accuracy and identifies where models excel and fail in specific reasoning stages, such as understanding relational semantic context, transferring relational learnings from one concept to another, and aligning modalities between text and vision. It provides insights for researchers to improve the limitations of the models.
> > > >
> > > > The low-performance results highlight the necessity for VOILA as a valuable and practical benchmark to push the boundaries of MLLMs, challenging them with complex reasoning tasks that demand human-like intelligence. As a large-scale, open-ended, and dynamic benchmark, VOILA offers flexibility to practitioners to select the desired properties and rule settings to generate visual analogy questions for different scenarios. The models can be trained on VOILA to develop new abilities or improve their shortcomings. With the decomposed tasks, VOILA not only evaluates the models' analogical reasoning abilities but also highlights specific shortcomings that must be addressed for the solution. With challenges and limitations, VOILA can encourage researchers to develop new architectures and algorithms to solve high-level analogy reasoning problems.
> > > >
> > > > We believe that our response addresses your concerns comprehensively and encourage you to reevaluate our submission positively. We look forward to further discussion.
> > > >
> > > > ---
> > > > [9] Zhou, D., Scharli, N., Hou, L., Wei, J., Scales, N., Wang, X., Schuurmans, D., Bousquet, O., Le, Q., & Chi, E.H.(2022). Least-to-Most Prompting Enables Complex Reasoning in Large Language Models.ArXiv, abs/2205.10625.

---

> > > > > ### Comment · Reviewer_qyEy · 2024-11-27
> > > > >
> > > > > Thank the authors for their responses! Their answers to the first question addressed the reviewer's concerns, and it'd be great to include this discussion in the paper. Regarding the benchmark's practical value, the authors made some good points but the reviewer would recommend adding more nuanced discussion on using this dataset for training models as it consists of synthetic images. The reviewer has raised their score.

---

> > > > > > ### Author Response · Authors · 2024-11-27
> > > > > > **Official Comments by Authors**
> > > > > >
> > > > > > Thank you for your thoughtful feedback and for raising your score. We sincerely appreciate your recognition of our efforts to address the major concerns. We will thoroughly revise the paper to incorporate all of your feedback.

---

### Official Review · Reviewer_Jyff · 2024-11-04

**Soundness:** 3
**Presentation:** 3
**Contribution:** 2
**Rating:** 6
**Confidence:** 4

**Summary:**

This paper proposed a new benchmark named VOILA, aiming to evaluate the performance of visual language models in understanding abstract relation understanding by designing visual analogy questions.
Experiments show that many models can describe the images and idenitify the relationship between images, but cannot apply the relationship very reliably.

**Strengths:**

1. The visual analogy perspective of evaluating vision language models is interesting.
2. The paper gives a very detailed the description of how the benchmark is collected.

**Weaknesses:**

1. The analogies considered in this paper seems to be restricted to only numbers, subjects, and actions.

**Questions:**

1. One line of related works I think this paper need to mention is on the benchmarking of multiple image understanding, such as works of Muirbench and MIRB, especially MIRB, which has already included visual analogy as one sub task for testing current state-of-the-art visual language models.
2. The data generation pipeline uses diffusion models to geerate images from given text prompts. I'm wondering how reliable is the diffusion model? how many images are discarded. Perhaps include a performance of human on the final dataset can answer this (I noticed that in table 3 there are some human performance, but why is human performance only tested on the 'Applying relationship' subset?)
3. In my understanding, this paper only considers analogy on properties like number, subject, and action. Is these enough to cover all possible visual combinations that can form analogies?



Muirbench: https://arxiv.org/abs/2406.09411
MIRB: https://arxiv.org/abs/2406.12742

---

> ### Author Response · Authors · 2024-11-19
> **Response to Reviewer Jyff**
>
> We thank the reviewer for the insightful comments on our paper. We are pleased to see that our work, VOILA, is recognized as **interesting**, with the benchmark collection method **detailedly explained.**
>
> Please find our detailed responses to your specific inquiries below:
>
> **“One line of related works I think this paper need to mention is on the benchmarking of multiple image understanding, such as works of Muirbench and MIRB, especially MIRB, which has already included visual analogy as one subtask.”**
>
> Although the MIRB [4] benchmark includes visual analogy as one of the sub-tasks, the authors acknowledged that their visual analogy task is **taken from the VASR [5] dataset which we already discussed in our Related Work section.**
>
> The commonality of MUIRBENCH [5] and VOILA is to investigate the relations on multi-image inputs. However, MUIRBENCH focuses on 12 diverse multi-image understanding categories like a diagram, geographic, scene, cartoon, etc., and 10 relation types like narrative, temporal, independent, cropped/zoomed, etc. On the other hand, VOILA concentrates on arithmetic and pattern understanding using numbers, subjects, and actions and rule-based relation identification and application on multiple images. The comprehensive scope of MUIRBENCH does not cover the visual analogy task we implemented in VOILA, but we will discuss the commonality of the multiple-image understanding and visual analogical reasoning in the final draft.
>
> **“How reliable is the diffusion model used in the data generation pipeline, and how many images are discarded?”**
>
> As described in section 3.1 Data Cleaning, 30 images per prompt were generated by implementing the Stable Diffusion XL model. To verify the quality of the images and their alignments with provided text prompts, they were manually filtered and 10 images for each prompt and a total of 7280 diverse images were used to create visual analogy questions. Since all the images were cleaned and validated by human participants during the dataset creation process, the images in analogy questions are reliable and do not require additional human quality checks.
>
> **“Why is human performance only tested on the "Applying relationship" step?”**
>
> After obtaining unsatisfactory results from MLLMs applying a direct answering (one-step) approach, we adopted L2M methods with multiple steps to reveal the shortcomings sub-tasks of the models. However, the human performance was conducted where participants were tasked with predicting the properties of the missing fourth image with one step, which resulted in satisfactory outcomes. We did not see the need for additional experiment to evaluate their performance on previous steps. Since they do not perform intermediate stages and directly answer the analogy questions with text format, their performance is displayed in the Applying Relationship step in Table 3.
>
> **“Is using properties like number, subject, and action enough to cover all possible visual combinations that can form analogies?”**
>
> VOILA was designed as a foundational framework, the limited count of properties was provided for the initial evaluation process. The utilized properties may not cover all possible visual combinations; however, their diversity is adequate to assess the state-of-the-art MLLMs' reasoning capability, as detailed experiment results show. Extending the variation of properties would likely create a more comprehensive benchmark, however, it would increase the complexity of the task which current MLLMs already suffer. We preferred at this stage not to overwhelm the models with the rising number of properties, since the current visual analogy configuration already challenges them, but leaving it as future scope after models evolve.
>
> We believe our response addresses your concerns comprehensively and encourage you to reevaluate our submission. We look forward to further discussion.
>
> ---
> [4] Zhao, B., Zong, Y., Zhang, L., & Hospedales, T. (2024). Benchmarking multi-image understanding in vision and language models: Perception, knowledge, reasoning, and multi-hop reasoning. arXiv. https://arxiv.org/abs/2406.12742
>
> [5] Bitton, Yonatan et al. “VASR: Visual Analogies of Situation Recognition.” AAAI Conference on Artificial Intelligence (2022).
>
> [6] Wang, F., Fu, X., Huang, J. Y., Li, Z., Liu, Q., Liu, X., Ma, M. D., Xu, N., Zhou, W., Zhang, K., Yan, T. L., Mo, W. J., Liu, H.-H., Lu, P., Li, C., Xiao, C., Chang, K.-W., Roth, D., Zhang, S., Poon, H., & Chen, M. (2024). MuirBench: A comprehensive benchmark for robust multi-image understanding. arXiv. https://arxiv.org/abs/2406.09411

---

> > ### Comment · Reviewer_Jyff · 2024-11-27
> >
> > Thanks for the rebuttal, it has cleared my concerns.
> > Please make sure to cite all the related works mentioned in the rebuttal in the final paper.
> > I would like to raise my score.

---

> > > ### Author Response · Authors · 2024-11-27
> > > **Official Response by Authors**
> > >
> > > Thank you very much for your thoughtful feedback and for raising your score. We sincerely appreciate your recognition of our efforts to address the major concerns. We will thoroughly revise the final paper to incorporate your feedback.

---

### Official Review · Reviewer_YkdC · 2024-11-04

**Soundness:** 2
**Presentation:** 3
**Contribution:** 2
**Rating:** 6
**Confidence:** 3

**Summary:**

The authors propose a new benchmark called VOILA which evaluates the perceptual understanding and analogical reasoning capabilities of multimodal language models, particularly its reasoning capabilities across multiple images. VOILA requires a MLLM to understand relations between images and generate a new image that follows the pattern. This task is open-ended and results are evaluated using a strong LLM. Results indicate that even the best MLLM performs much worse than an average human (30% vs 70%)

**Strengths:**

This is generally a well organized and well written paper.
The proposed benchmark is interesting and highlights a critical shortcoming in existing MLLMs
The benchmark is well curated, there is also manual filtering involved to ensure quality

**Weaknesses:**

This task is not exactly novel, [1] propose a new task that evaluates visual cognition of MLLMs. The new novelty comes from the requirement that MLLM generate the output image.
The evaluation is not grounded in existing psychological assessments for instance the reasoning used in [1] is widely used in neurodevelopmental and neuropsychological research.
GPT4o seems to be poor at identifying relationships (from image) in VOILA, does this affect the evaluation which in turn also uses GPT4o.

References
[1] https://arxiv.org/abs/2406.10424

**Questions:**

Can the authors conduct a small experiment to validate that their evaluation strategy is accurate?
The prompt to identify relation asks the model to identify differences in subject, types and actions all in a single prompt. What if this problem was broken down to three sub-prompts? It seems intuitive that performance should improve

---

> ### Author Response · Authors · 2024-11-20
> **Response to Reviewer YkdC (part 1)**
>
> We thank the reviewer for the insightful suggestions on our paper. We are pleased that you found VOILA **interesting, well-curated**, and designed with **manual filtering to ensure quality.** We are also pleased that you recognize that **VOILA highlights the critical weaknesses in state-of-the-art MLLMs.** Below, we provide additional clarifications and experiment results as requested:
>
> **“Novelty of the VOILA and comparison with MaRs-VQA benchmark”**
>
> We believe VOILA and the MaRs-VQA [1] benchmark are sufficiently distinct and thus VOILA has novelty. We agree that both studies work on abstract visual reasoning; however, MaRs-VQA are sourced from MaRs-IB [2] matrix reasoning questions, while VOILA has been **generated from scratch with novel rule-based configurations with the Analogy Building Algorithm** and the Stable Diffusion model. In addition to the dataset creation pipeline, the content, method, and structure of both studies are different.  While MaRs-IB consists of 1440 static questions created with synthetic abstract figures on 3 by-3 matrix structures with 10 VQA questions, VOILA which dynamically generates up to 6.4M questions, focuses on real image-based analogical reasoning utilizing 7280 manually cleaned images on A : B :: A’ :: B’ framework with a multi-step process. Additionally, the application of the MaRs-IB is established on multiple answer choices which causes 25% success in a random selection. However, as stated in the Introduction part of the paper, "According to Bloom’s taxonomy of educational objectives, creation, rather than evaluation, requires the highest cognitive skills in the learning process [3].”  By following this idea, VOILA implements **generating a solution approach rather than selection to measure the high-level reasoning skills of the models.**
>
> **“The evaluation of Voila is not grounded in existing psychological assessments for instance MaRs-VQA benchmark”**
>
> We respectfully disagree with the requirement of the psychological assessment in the VOILA benchmark. MaRs-VQA [1] is sourced from MaRs-IB matrix reasoning questions [2] which aim to investigate the developmental differences in reasoning capacity from adolescents to adults. Since MaRs-IB is conducted on a psychological baseline, the authors use neurodevelopmental and neuropsychological evaluations to validate the study. However, VOILA focuses on evaluating MLLMs’ high-level abstract visual reasoning capabilities and for this purpose, it utilizes analogical reasoning tasks which are widely used in IQ tests and cognitive assessments.
>
> **“What would happen if the task of identifying differences in subject, types, and actions in a single prompt was broken down into three separate sub-prompts?”**
>
> This is an interesting experiment that we did not investigate. As requested, we conducted a new experiment with GPT-4o on the VOILA-WD dataset to discover the impact of using subprompts on identifying property relationships. For a fair comparison, we froze the first-step answers and requested the model to find whether the number/subject/action changed or remained the same from the first image to the second image. After merging the results from three subtasks, we achieved a similar accuracy of 42% with a single prompt experiment. The accuracy of properties is also similar with 94% for numbers, 79% for subjects, and 56% for action. The relationship identification performance of GPT-4o with single vs three sub-prompts is provided in the Table below. The results demonstrate that **GPT-4o's ability to identify relationships is not affected by the number of properties asked in the prompt.** We will include the results of this ablation study in the final draft.
>
> Method          | Approach    | Number         | Subject         |       Action    |    Total     |
> ------------------|------------------|-------------------|------------------|------------------|--------------|
>  **GPT-4o** | Single Prompt   | **94.3%**    |  78.5%        | 55.9%       |  **42.8%**   |
>  **GPT-4o** | Three Sub-prompts | 94.1%     |  **79.3%**    | **56.3%**  |  42.3%     |
>
> ---
> [1] Cao, X., Lai, B., Ye, W., Ma, Y., Heintz, J., Chen, J., Cao, J., & Rehg, J. M. (2024). What is the visual cognition gap between humans and multimodal LLMs?. arXiv. https://arxiv.org/abs/2406.10424
>
> [2] Gabriele Chierchia, Delia Fuhrmann, Lisa J Knoll, Blanca Piera Pi-Sunyer, Ashok L Sakhardande, and Sarah-Jayne Blakemore. The matrix reasoning item bank (mars-ib): novel, 12open-access abstract reasoning items for adolescents and adults. Royal Society open science, 6(10):190232, 2019.
>
> [3] B. S. Bloom, M. B. Engelhart, E. J. Furst, W. H. Hill, and D. R. Krathwohl. Taxonomy of educational objectives. The classification of educational goals. Handbook 1: Cognitive domain. Longmans Green, New York, 1956.

---

> ### Author Response · Authors · 2024-11-20
> **Response to Reviewer YkdC (part 2)**
>
> **“Does GPT-4's difficulty in identifying relationships in VOILA affect the evaluation process, which also uses GPT-4?”**
>
> Thank you for raising this important question for discussion. Although the task of identifying relationships is different from comparing ground truths with model responses, the authors share the same wonder about how well GPT-4o performs in the evaluation task. To answer the question and measure the gap between human and GPT-4 evaluations, we conducted an error analysis, as requested. Since every model obtains distinct characteristic response styles, we evaluated **50 visual analogy question** responses from diverse models with various rule configurations including the distraction rule. After processing the intermediate stages, a total of **180 responses** were evaluated, with **30 corresponding to the image generation** step. For a comprehensive evaluation, we utilized confusion tables for each step including all attributes. The terminology and tables are given below.  The “Question” in tables represents the question answer merged with three properties (number + subject + action).
> - **True Positive (TP):** The number of cases where both the human and the GPT-4o agree that the response is correct.
> - **False Negative (FN):** The number of cases where the human expresses the answer is correct, but the GPT-4o says it is incorrect.
> - **False Positive (FP):** The number of cases where the human says the answer is incorrect, but the GPT-4o states it is correct.
> - **True Negative (TN):** The number of cases where both the human and the GPT-4o agree that the response is incorrect.
> - **Accuracy (Agreement):** The number of cases where both the human and the GPT-4o agree.
>    - Accuracy (Agreement rate) = (TP + TN) / (TP + TN + FP + FN)
>
> First, we calculated false negative and false positive cases between human and GPT-4o evaluations for each step, attribute, and question answer. Then we calculated the accuracy also called the agreement rate. The results show that out of 50 evaluated questions and their intermediate steps, the agreement rate between human evaluation and GPT-4o was **91% for describing images, 94% for identifying relations, 92% for applying relationships, and 91% for image generation.** Although the agreement rate for attributes is the lowest at 74%, the accuracy of question answers, which involve merging properties, exceeds 91%. This demonstrates that GPT-4o has a **maximum error rate of 10% per step**, and its difficulty in identifying relationships does **not affect the benchmark evaluation process.** We will add this comprehensive error analysis to the final draft.
>
> ### Step 1 - Describing Images
> | Attribute | FP | FN | TP + TN| Accuracy (Agreement)
> |-------------------|---------------|----------------|---------------|------------------|
> | **Number**   | 1 | 7 | 142 | 95%
> | **Subject** | 5 | 5 |140 | 93%
>  | **Action** | 5 | 14 |131 | 87%
>  | **Question** | 6 | 7| 137 |**91%**
>
> ### Step 2 - Identifying Relations
> | Attribute | FP | FN | TP + TN| Accuracy (Agreement)
> |-------------------|---------------|----------------|----------------|-------------|
> | **Number**   | 4 | 5 | 39 |78%
> | **Subject** | 1 | 12 |37 |74%
>  | **Action** | 2  | 8  | 40 |80%
>  | **Question** | 1 | 2| 47 |**94%**
>
> ### Step 3 - Applying Relationships
> | Attribute | FP | FN | TP + TN| Accuracy (Agreement)
> |-------------------|---------------|----------------|-----------|---------------|
> | **Number**   | 3 | 8 | 39 | 78%
> | **Subject** | 4 | 7 |39 | 78%
>  | **Action** | 8  | 4  | 38 | 76%
>  | **Question** | 2 | 2| 46 | **92%**
>
> ### Step 4 - Generating Images
> | Attribute | FP | FN | TP + TN| Accuracy (Agreement)
> |-------------------|---------------|----------------|-------------|---------------|
> | **Number**   | 5  | 2 | 23 |77%
> | **Subject** | 1 | 4  | 25 |83%
>  | **Action** | 2  | 4  | 24 |80%
>  | **Question** | 1 | 2 | 27 |**90%**
>
> We believe our response and new experiments address your concerns comprehensively and encourage you to reevaluate our submission. We look forward to further discussion.

---

> ### Comment · Reviewer_YkdC · 2024-11-26
> **Response**
>
> Dear Authors,
>
> Thank You for the detailed responses and the new experiments. It is good to see that GPT-4's difficulty in identifying relationships does not affect VOILA. Thank You for the comparison with MaRs-VQA benchmark. I am raising my score to 6 in light of the clarified concerns and helping me understand the soundness of the paper. I am not raising my score further as the idea or implementation is not radically new but a good follow up work to an established line of works . Thank You.

---

> > ### Author Response · Authors · 2024-11-27
> > **Official Response by Authors**
> >
> > Thank you for your thoughtful feedback and for raising your score. Your suggestions are valuable in helping us improve the paper. We will thoroughly revise the paper to include ablation studies.

---

### Official Review · Reviewer_pCZi · 2024-11-05

**Soundness:** 3
**Presentation:** 3
**Contribution:** 3
**Rating:** 6
**Confidence:** 3

**Summary:**

The paper introduces VOILA, a benchmark specifically designed to evaluate Multimodal Large Language Models (MLLMs) in perceptual understanding and abstract relational reasoning across images. By using analogical mapping, VOILA requires MLLMs to generate an image that completes an analogy between two image pairs, testing the models' relational reasoning without predefined answer choices. The benchmark includes challenging tasks, with models struggling to match human performance, especially in higher-level reasoning steps. Performance improves with least-to-most prompting strategies, but there remains a substantial gap between the best-performing model and human results.

**Strengths:**

• VOILA stands out by focusing on abstract visual analogies, making it a valuable addition to existing MLLM evaluation benchmarks, particularly in assessing perceptual and relational reasoning.

• The multi-step reasoning approach and comprehensive ablation studies offer a detailed examination of the current limitations in MLLMs.

**Weaknesses:**

• The paper emphasizes that this is a dynamic benchmark. For a static benchmark, once a configuration is provided, it can be fully generated and then annotated for correctness by human evaluators. However, as a dynamic benchmark, once it is generated, how to ensure the correctness of the benchmark and how to make it scalable, which remain challenging and difficult to guarantee. This issue is unsolved in this paper.

• VOILA’s reliance on GPT-4o for model evaluation makes the evaluation process both resource-intensive and costly for practitioners.

**Questions:**

Please refer to Weaknesses

---

> ### Author Response · Authors · 2024-11-19
> **Response to Reviewer pCZi**
>
> We are encouraged by your positive feedback.  We are pleased that you recognize **VOILA reveals the limitations of state-of-the-art MLLMs with comprehensive ablation studies and detailed examinations.** Below, we provide additional clarifications on the accuracy and scalability of the VOILA dynamic benchmark and resource usage of GPT-4o as requested:
>
> **“How to ensure the correctness of dynamic benchmark and how to make it scalable?”**
>
> The correctness of the dynamic VOILA dataset is ensured by a **rigorous manual cleaning** process of the 7280 images generated with the Stable Diffusion XL model based on pre-defined properties. Since all cleaned images include correct property tags, practitioners can build visual analogy questions running the Visual Analogy Generation algorithm without any annotation requirements.
>
> VOILA offers the practitioners flexibility to work on various properties and rule configurations from the predefined set. The practitioner can modify the dataset size and select the desired properties and rule settings to generate visual analogy questions for different scenarios. The Visual Analogy Generation algorithm builds questions with a balanced distribution of image use, properties, and rules. Instead of a fixed dataset limit, the **algorithm dynamically creates over 6.4 million analogy questions which makes VOILA highly scalable.**
>
> **“GPT-4o makes the evaluation process resource-intensive and costly”**
>
> We respectfully disagree with the resource-intensive evaluation process. To optimize our evaluation pipeline, we compared the open-source Llama-3-70B-Instruct model with GPT-4o. While **Llama-3-70B-Instruct requires more than 12 hours on 2 A100 GPUs** to evaluate 10K results of a single model **for each step**, using Batch API of **GPT-4o reduces the time to approximately 45 minutes for entire three steps** with parallel processing without any GPU requirements.
>
> The cost of GPT-4o API is calculated based on input and output tokens. The cost and the result time of the evaluation pipeline vary depending on the length of the models’ response. In our experiments, the API uses roughly 450, 300, and 230, 60 input tokens and precisely 74, 23, 24, and 30 output tokens sequentially at each step. Evaluating 10K questions costs *$13.75* for input tokens and *$9.05* to generate output. In total, the evaluation cost of the entire task, including the image generation step for a single model is **approximately $22.80**. Suppose the practitioners do not prefer to follow the multi-step process. In that case, they can evaluate the text-based results in the third step with a Python script by matching the provided ground truth answers with the models’ responses. Although GPT-4o requires payment, it obtains several benefits to optimize the evaluation pipeline, including a faster process, high resource efficiency, easy result analysis in JSON format, and good quality accuracy.
>
> We believe these clarifications address all your concerns and questions. We look forward to further discussion if needed.

---

> > ### Author Response · Authors · 2024-12-01
> > **Follow-Up on Discussion Response**
> >
> > We sincerely appreciate your thoughtful feedback on our paper and are greatly encouraged by the positive insights you have provided. We have provided detailed explanations to address your concerns about dynamic benchmark and GPT-4o evaluation. As the discussion is nearing its conclusion, we wanted to kindly confirm if our responses have adequately addressed all your concerns or if any remaining points require further clarification. Thank you for your dedicated time and effort in this review process.

---

### Author Response · Authors · 2024-11-23
**Global Response to Reviewers**

We greatly appreciate the valuable and constructive feedback provided by the reviewers. We are truly gratified to see their unanimous positive evaluations, which acknowledge the strengths of our work across multiple dimensions.

- The reviewers consistently acknowledge that **VOILA reveals the shortcomings of state-of-the-art MLLMs** (pCZi, YkdC, pi5r) supported by **detailed and comprehensive evaluations**. (pCZi, qyEy, pi5r).
- Reviewers pCZi and pi5r recognize the significance of VOILA in **contributing valuable advancements to abstract relational reasoning** within MLLMs.
- Reviewers qyEy and pi5r acknowledge VOILA’s **contribution to the dataset creation pipeline**, noting its **well-defined and comprehensive structure**, which includes clear definitions of various properties, rule configurations, and difficulty levels.
- Reviewer Jyff emphasizes the critical stage of the **well-curated** VOILA task —**applying relationships**—which highlights the distinction between human reasoning ability and MLLMs.
- Additionally, we are pleased that reviewer qyEy described our study as **unique, well-motivated, and well-formulated**, and Reviewer pi5r acknowledged VOILA task as a **novel multimodal reasoning problem**.

We have addressed each reviewer’s comments in detail individually. Below, we summarize our responses to the key questions and offer additional clarifications.

## Clarification of the Significance of Analogical Reasoning and VOILA
Analogical reasoning is a critical cognitive ability for effective problem-solving, decision-making, and learning strategies. A key aspect of analogical reasoning is the transfer of knowledge from previous relational learnings to novel concepts, which distinguishes it from other reasoning tasks. By identifying similarities between past experiences and new situations and applying relational patterns, humans enhance their learning process, solve problems more effectively, and anticipate the outcomes of their decisions.

In this work, we created VOILA, an analogical reasoning task, to investigate how well current MLLMs can transfer learned patterns to new domains, essential for generalization. By evaluating the models' performance on VOILA, supported by comprehensive analysis, we aim to reveal their limitations and demonstrate the gap in analogical reasoning abilities between humans and MLLMs.

## Summary of Ablation Studies
- **Ablation on the Accuracy of GPT-4o in the Evaluation Process** : As requested by Reviewer YkdC, we conducted a thorough comparison of human performance and GPT-4o performance across evaluation steps and properties. Our findings indicate that GPT-4 achieves high accuracy in comparing and scoring ground truth and model answers, with a maximum error rate of only 10%.

- **Ablation on Relation Extraction Step Using Three Sub-prompts** : In response to Reviewer YkdC's request, we conducted an additional ablation study by using three sub-prompts instead of one to reveal the relationships between properties in images. We observed that GPT-4o's ability to identify relationships remains unaffected by the number of properties requested in the prompt.

- **Ablation on the AnyRes Method for Image Collages** : As requested by Reviewer qyEy, we performed additional ablations on Llava-One-Vision model using AnyRes approach, processing both image collages and three sequential images as input. Our findings indicate that the AnyRes method enhances single-image resolution and narrows the performance gap between processing image collages and multiple separate images.

- **Ablation on Performance Gap Between Human and MLLMs**: In response to Reviewer pi5r, we conducted an additional ablation study with GPT-4o on VOILA-WD under the same human evaluation conditions, incorporating the direct answering approach. The model was provided with two example questions and property option lists. We observed that the new setting did not increase the performance of MLLLMs, as compared to our previous study using the least-to-most (L2M) approach with step-by-step instructions.

- **Ablation on CoT Effect at "Applying Relationship" Step**: In response to Reviewer pi5r, we conducted an ablation study with GPT-4o on VOILA-WD to test the impact of the CoT approach at the third step. We provided two textual examples and rationales for the model. Our findings indicate that the CoT approach performs similarly to the L2M method which implements a multi-step reasoning process with detailed instructions.

Once again, we thank the reviewers for their time and detailed feedback. We are glad that we resolved most of the queries from the reviewers. We believe that this fruitful discussion has made our paper stronger.

---

### Meta-Review · Area_Chair_onb4 · 2024-12-05

**Metareview:**

### Summary of Scientific Claims and Findings
This paper introduces **VOILA**, a benchmark for evaluating Multimodal Large Language Models (MLLMs) on their ability to perform abstract relational reasoning and perceptual understanding through visual analogies. VOILA presents a novel problem structure that integrates dynamic, large-scale datasets with varying levels of complexity (VOILA-ND and VOILA-WD). The key task requires MLLMs to generate images or textual outputs that complete an analogy based on provided image pairs, testing models' ability to identify, understand, and apply relationships between properties such as numbers, subjects, and actions. Experimental results demonstrate that while current MLLMs can perform simple tasks such as property identification, they struggle significantly with higher-order relational reasoning, especially when distraction rules are introduced.

### Strengths of the Paper
1. **Novel Contribution**: VOILA introduces a unique problem structure, focusing on abstract analogical reasoning, which is distinct from existing benchmarks like MaRs-VQA and MIRB.
2. **Comprehensive Benchmark**: The dynamic dataset generation process, covering over 6.4M analogy questions, makes VOILA scalable and adaptable to different configurations.
3. **Thorough Evaluation**: Extensive experiments reveal the limitations of current MLLMs in high-level reasoning tasks, with detailed ablation studies exploring various task aspects.
4. **Impactful Insights**: The paper highlights significant gaps in MLLMs’ reasoning capabilities, motivating further research in multimodal reasoning.

### Weaknesses and Missing Elements
1. **Practical Significance**: While the benchmark is well-defined, its practical applications and relevance to real-world scenarios could be better articulated.
2. **Evaluation Noise**: The use of external models like DALL-E 3 for image generation introduces potential noise in evaluating reasoning capabilities.
3. **Presentation Gaps**: Dense tables and some unclear explanations limit accessibility. Insights from results could be highlighted more prominently.
4. **Limited Novelty in Methods**: Although the problem is novel, the methods (e.g., Least-to-Most prompting) align with prior work, making the approach feel incremental.
5. **Image Generation Challenges**: There is limited error analysis explaining why models fail at the image generation stage and how this step contributes uniquely to reasoning evaluation.

### Recommendation and Justification
The paper makes a significant contribution by introducing VOILA, a novel benchmark for evaluating multimodal large language models (MLLMs) on abstract analogical reasoning and perceptual understanding, a largely unexplored area in current research. VOILA offers a scalable, dynamic dataset and reveals critical gaps in MLLMs' reasoning capabilities compared to humans, supported by robust experiments and detailed ablation studies. The authors effectively addressed most reviewer concerns, including the benchmark’s novelty, relevance, and evaluation methodologies, with additional experiments and clarifications. While some issues remain, such as evaluation noise in the image generation step and limited practical applications discussed, these do not diminish the paper’s overall merit. Given its potential to drive future advancements in multimodal reasoning and its alignment with reviewer consensus post-rebuttal, this work is a valuable addition to the field.

**Additional Comments On Reviewer Discussion:**

### Key Discussion Points and Author Responses:
1. **Novelty and Benchmark Scope**: Some reviewers questioned the novelty of VOILA compared to existing benchmarks like MaRs-VQA. The authors clarified that VOILA dynamically generates analogies with new rule configurations, focusing on application and abstraction, which distinguishes it from static datasets.
2. **Image Generation Relevance**: Concerns about the necessity and noise in the image generation step were raised. The authors justified this step as crucial for multimodal reasoning evaluation, highlighting distinct challenges in translating relational reasoning into visual outputs.
3. **Human vs. Model Comparisons**: Reviewers noted potential unfairness in comparing human and model performance. The authors demonstrated that even when providing models with options and examples, they significantly underperform relative to humans.
4. **Prompting Strategies**: Some reviewers suggested implementing Chain-of-Thought (CoT) reasoning for better performance. The authors provided ablation studies showing that Least-to-Most prompting yields similar or better results than CoT for this task.
5. **Practical Relevance**: Questions about the real-world applicability of VOILA were addressed with examples like autonomous driving and medical diagnostics, where relational reasoning is critical.

### Final Decision Weighing:
The authors addressed most reviewer concerns comprehensively, adding valuable ablation studies and clarifications. Remaining issues, such as evaluation noise and practical significance, do not detract from the benchmark’s core contributions. The overall improvements during the rebuttal period support the paper’s acceptance as a meaningful and impactful contribution to the field.

---

### Decision · Program_Chairs · 2025-01-22

Accept (Poster)